# Touch receptor end-organ innervation and function require sensory neuron expression of the transcription factor Meis2

**Simon Desiderio[1†], Frederick Schwaller[2†], Kevin Tartour[3], Kiran Padmanabhan[3], Gary R Lewin[2], Patrick Carroll[1], Frederic Marmigere[3]\***

[1]Institute for Neurosciences of Montpellier, University of Montpellier, INSERM U 1298, Montpellier, France; [2]Department of Neuroscience, Max-Delbrück Centre for Molecular Medicine, Berlin-Buch, Germany; [3]IGFL, UMR 5242 CNRS/ENS, Lyon, France

**\*For correspondence:**
frederic.marmigere@ens-lyon.fr

[†]These authors contributed equally to this work

**Competing interest:** The authors declare that no competing interests exist.

**Abstract** Touch sensation is primarily encoded by mechanoreceptors, called low-threshold mechanoreceptors (LTMRs), with their cell bodies in the dorsal root ganglia. Because of their great diversity in terms of molecular signature, terminal endings morphology, and electrophysiological properties, mirroring the complexity of tactile experience, LTMRs are a model of choice to study the molecular cues differentially controlling neuronal diversification. While the transcriptional codes that define different LTMR subtypes have been extensively studied, the molecular players that participate in their late maturation and in particular in the striking diversity of their end-organ morphological specialization are largely unknown. Here we identified the TALE homeodomain transcription factor Meis2 as a key regulator of LTMRs target-field innervation in mice. *Meis2* is specifically expressed in cutaneous LTMRs, and its expression depends on target-derived signals. While LTMRs lacking *Meis2* survived and are normally specified, their end-organ innervations, electrophysiological properties, and transcriptome are differentially and markedly affected, resulting in impaired sensory-evoked behavioral responses. These data establish *Meis2* as a major transcriptional regulator controlling the orderly formation of sensory neurons innervating peripheral end organs required for light touch.

## eLife assessment

This **fundamental** study identifies the homeodomain transcription factor Meis2 as a transcriptional regulator of maturation and end-organ innervation of low-threshold mechanoreceptors (LTMRs) in the dorsal root ganglia (DRG) of mice. The authors use histology, behavioral tests, RNA-sequencing, and electrophysiological recordings to provide evidence that conditional deletion of Meis2 in post-mitotic DRG neurons causes gene expression changes together with targeting errors and altered sensory neuron responses, ultimately resulting in reduced sensitivity to light touch in mutant animals. The data presented are **convincing**, the discussion comprehensive, and the conclusions drawn justified.

## Introduction

Tactile stimuli like brush, light pressure, or roughness engage highly specialized and diverse arrays of mechanoreceptors in both the hairy and glabrous skin (*Abraira and Ginty, 2013*; *Delmas et al., 2011*; *Handler and Ginty, 2021*; *Lechner and Lewin, 2013*; *Wu et al., 2021*; *Zimmerman et al., 2014*;

*Li et al., 2011*). Somatosensory perception via these mechanoreceptors involves primary sensory neurons whose cell bodies reside within dorsal root ganglia (DRG) and cranial sensory ganglia. These sensory neurons within the DRG can be broadly classified as nociceptors, mechanoreceptors, or proprioceptors, and each group is characterized by the expression of specific combination of genes and have distinctive physiological properties and projections within the spinal cord and periphery (*Lallemend and Ernfors, 2012*; *Marmigère and Ernfors, 2007*; *Vermeiren et al., 2020*).

Cutaneous mechanoreceptors or low threshold mechanoreceptors (LMTRs) exhibit a variety of specialized terminal endings in the hairy and glabrous skin with strikingly unique morphologies (*Abraira and Ginty, 2013*; *Delmas et al., 2011*; *Handler and Ginty, 2021*; *Lechner and Lewin, 2013*; *Zimmerman et al., 2014*; *Li et al., 2011*; *Sharma et al., 2020*; *Schwaller et al., 2021*). LTMRs projecting to the glabrous skin innervate Merkel cell complexes or Meissner corpuscles at the dermal–epidermal border. Those innervating Merkel cells in the glabrous or hairy skin have large thickly myelinated axons (Aβ-fibers) and are characterized as slowly adapting mechanoreceptors responding to skin movement and static displacement (also referred to as Aβ-SAIs). On the other hand, Meissner corpuscles are mechanoreceptors which are only sensitive to skin movement or vibration (rapidly adapting mechanoreceptors) and are referred to as Aβ-RAs. LTMRs innervating hair follicles in the hairy skin can form lanceolate endings or circumferential endings. Virtually all mechanoreceptors innervating hairs show rapidly adapting properties and respond only to hair movement, but not to static displacement (*Lechner and Lewin, 2013*). LTMRs with large myelinated axons innervating hairy skin are characterized as Aβ-RAs, and a specialized population of slowly conducting myelinated fibers called D-hair mechanoreceptors (or Aδ-RAs) also form lanceolate endings on small hairs. D-hair mechanoreceptors are most sensitive to low-velocity stroking, have large receptive fields, and are directionally tuned (*Li et al., 2011*; *Walcher et al., 2018*). A small number of LTMRs in the hairy skin are not activated by hair movement but show properties of rapidly adapting mechanoreceptors (*Lewin and McMahon, 1991*). These were originally characterized as so-called field receptors (*Lewin and McMahon, 1991*; *Burgess and Horch, 1973*) and were recently shown to form circumferential endings around hair follicles (*Bai et al., 2015*). LTMRs tuned to high-frequency vibration are called Aβ-RAII and innervate Pacinian corpuscles deep in the skin or on the bone (*Schwaller et al., 2021*). Ruffini endings that are thought to be innervated by stretch-sensitive mechanoreceptors (Aβ-SAII) remain poorly characterized in mice (*Handler and Ginty, 2021*). Recent advances in combining single-cell transcriptomic and deep RNA sequencing with genetic tracing have tremendously extended the classical subtypes repertoire and clustered at least 20 different subtypes of LTM neurons (*Wu et al., 2021*; *Sharma et al., 2020*; *Bai et al., 2015*; *Usoskin et al., 2015*; *Zheng et al., 2019*).

Cracking the transcriptional codes supporting sensory neuron identity and diversification has been the object of tremendous efforts in the last decades (*Wu et al., 2021*; *Lallemend and Ernfors, 2012*; *Marmigère and Ernfors, 2007*; *Sharma et al., 2020*; *Usoskin et al., 2015*; *Zheng et al., 2019*). For instance, the functions of specification factors or terminal selectors, like Maf, Shox2, Runx3, Pea3, and ER81, have been functionally implicated in LTMR segregation (*Lallemend and Ernfors, 2012*; *Marmigère and Ernfors, 2007*; *Scott et al., 2011*; *Abdo et al., 2011*; *Arber et al., 2000*; *Hu et al., 2012*). Whereas the specific function of adhesion molecules in shaping the assembly of touch circuitry is being unraveled (*Meltzer et al., 2023*), the transcriptional control of target cell innervation within the skin and of the establishment of specialized peripheral end-organ complexes is less understood. Meis2 is another TF expressed in LTMRs (*Sharma et al., 2020*; *Usoskin et al., 2015*; *Zheng et al., 2019*). Its mutation in humans causes severe neurodevelopmental defects (*Gangfuß et al., 2021*; *Giliberti et al., 2020*; *Shimojima et al., 2017*), and somatic mutations of its DNA consensus binding site are associated with neurodevelopmental defects (*Bae et al., 2022*). It belongs to a highly conserved homeodomain family containing three members in mammals, Meis1, Meis2, and Meis3 (*Geerts et al., 2003*; *Longobardi et al., 2014*), and *Meis1* is necessary for target-field innervation of sympathetic peripheral neurons (*Bouilloux et al., 2016*). We thus wondered if Meis2 could also be a pertinent regulator of late primary sensory neuron differentiation.

Here, we show that Meis2 regulates the innervation of specialized cutaneous end organs important for LTMR function. We confirmed that *Meis2* expression is restricted to LTMR subclasses at late developmental stages compatible with functions in specification and/or target-field innervation. We generated mice carrying an inactive *Meis2* gene in postmitotic sensory neurons. While these animals are healthy and viable and do not exhibit any neuronal loss, they display tactile sensory defects in

electrophysiological and behavioral assays. Consistent with these findings, we identified major morphological alterations in LTMR end-organ structures in *Meis2* null sensory neurons. Finally, transcriptomic analysis at late embryonic stages showed dysregulation of synapse and neuronal projection-related genes that underpin these functional and behavioral phenotypes.

## Results

### *Meis2* is expressed by cutaneous LTMRs

We analyzed *Meis2* expression using in situ hybridization (ISH) at various developmental stages in both mouse and chick lumbar DRG, combined with well-established molecular markers of sensory neuron subclasses (*Figure 1A*, *Figure 1—figure supplements 1 and 2*). In mouse, *Meis2* mRNA was first detected at embryonic day (E) 11.5 in a restricted group of large DRG neurons. This restricted expression pattern was maintained at E14.5, E18.5, and adult stages (*Figure 1A*). In chick, *Meis2* was expressed in most DRG neurons at Hamburger–Hamilton stage (HH) 24, but later becomes restricted to a well-defined subpopulation in the ventrolateral part of the DRG where LTMRs and proprioceptors are located (*Rifkin et al., 2000*; *Figure 1—figure supplement 2A*). In both species, *Meis2*-positive cells also expressed the pan-neuronal marker Islet1, indicating that they are postmitotic neurons. In chick, we estimated that *Meis2*-positive cells represented about 15% of Islet1-positive DRG neurons at HH29 and HH36, respectively, suggesting a stable expression in given neuronal populations during embryonic development. Double ISH for *Meis2* and *Ntrk2*, *Ntrk3* or *Ret* mRNAs in E14.5 and E18.5 mouse embryonic DRG (*Figure 1—figure supplement 1A and B*) showed a large co-expression in *Ntrk2*- and *Ntrk3*-positive neurons confirming that *Meis2*-positive neurons belong to the LTMR and proprioceptive subpopulations. Finally, double ISH for *Meis2* and *Ret* in E14.5 mouse DRG showed that virtually all large *Ret*-positive neurons representing part of the LTMR pool co-expressed *Meis2* at this stage before the emergence of the small nociceptive *Ret*-positive population. Similar results were found in chick at HH29 (*Figure 1—figure supplement 2B*).

In mouse, comparison of *Meis2* mRNA expression to *Ntrk1*, a well-established marker for early nociceptive and thermo-sensitive neurons, showed that only a few *Meis2*-positive neurons co-expressed *Ntrk1* at E11.5 and E18.5 (*Figure 1—figure supplement 1C*). In chick HH29 embryos, *Meis2* expression was fully excluded from the *Ntrk1* subpopulation (*Figure 1—figure supplement 2C*). In adult mouse DRG, comparison of *Meis2* mRNA to Ntrk1, Calca, and TrpV1 immunostaining confirmed that *Meis2*-expressing neurons are largely excluded from the nociceptive and thermosensitive populations of DRG neurons. Instead, a large proportion of *Meis2*-positive neurons co-expressed *Nefh*, a marker for large myelinated neurons including LTMR and proprioceptors, and *Pvalb*, a specific marker for proprioceptors (*Figure 1—figure supplement 1D*). Finally, *Meis2* expression in LTMRs projecting to the skin was confirmed by retrograde-tracing experiments using cholera toxin B subunit (CTB) coupled with a fluorochrome injected into hind paw pads of P5 newborn mice. Analyses of CTB expression in lumbar DRG 3 d later at P8 showed that many retrogradely labeled sensory neurons were also immunopositive for Meis2, Maf, Ntrk2, and Ntrk3 (*Figure 1B*).

Altogether, our results on Meis2 co-localization with Nefh, Ntrk2, Ntrk3, Ret, Pvalb, and Maf at different embryonic and postnatal stages are consistent with previous reports on restricted Meis2 expression to the Aβ-field, Aβ-SA1 and Aβ-RA subclasses of LTMR neurons and proprioceptive neurons (*Sharma et al., 2020*; *Usoskin et al., 2015*; *Zheng et al., 2019*; *Shin et al., 2020*). The relatively lower coincidence of Meis2 and Ntrk2 expressions compared to Ntrk3 is consistent with Meis2 being excluded from the Aδ-LTMRs (D-hair mechanoreceptors). The lack of co-expression with Ntrk1 and TrpV1 also confirmed Meis2 exclusion from peptidergic and non-peptidergic subpopulations.

### Target-derived signals are necessary to maintain *Meis2* expression

The requirement for extrinsic signals provided by limb mesenchyme and muscles for proprioceptor and LTMR development has been documented (*Sharma et al., 2020*; *Arber et al., 2000*; *Shin et al., 2020*; *Patel et al., 2003*; *de Nooij et al., 2013*; *Poliak et al., 2016*; *Wang et al., 2019*). To test the influence of target-derived signals on *Meis2* expression in sensory neurons, limb buds were unilaterally ablated in HH18 chick embryos. Embryos were harvested at HH29 and HH36, before and after ventrolateral neurons are lost, respectively (*Oakley et al., 1995*; *Oakley et al., 1997*; *Calderó et al., 1998*; *Figure 1C and D*, *Figure 1—figure supplement 2D*). In HH36 embryos, about 65% of Meis2-positive

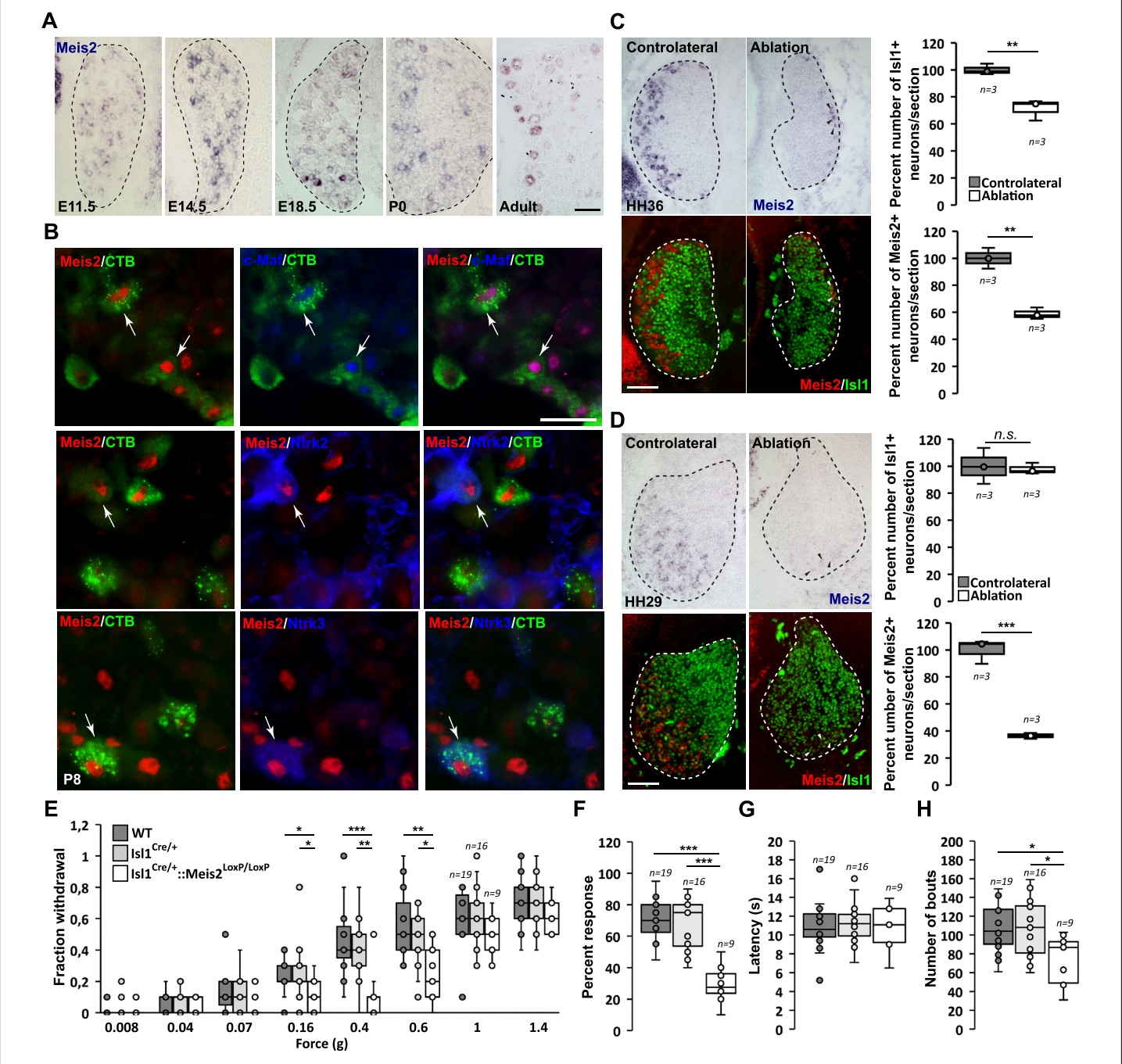

**Figure 1.** *Meis2* is expressed in subclasses of dorsal root ganglia (DRG) cutaneous mechanoreceptive neurons in mouse embryos. (**A**) In situ hybridization (ISH) for *Meis2* mRNA showed expression in a subpopulation of DRG sensory neurons at embryonic stages E11.5, E14.5, and E18.5, at P0, and at adult stages. Dashed lines delineate the DRG. Scale bar = 50 µm. (**B**) IF for Meis2 (red) and c-Maf, Ntrk2, or Ntrk3 (blue) at P7 following injection of cholera toxin B subunit (CTB in green) in the skin of newborn pups. Note that Meis2⁺/CTB⁺ retro-traced sensory neurons co-expressed c-Maf, Ntrk2, or Ntrk3 (arrows). Scale bar = 50 µm. We estimated that 30.5 ± 3.5% (mean ± SEM; n = 3) of Meis2-positive neurons co-expressed Ntrk2, and that 39.5 ± 5.4% co-expressed Ntrk3. Conversely, Meis2 was co-expressed in 53.6 ± 9.4% of Ntrk2-positive neurons, and in 78.5 ± 5.0% of Ntrk3-positive neurons. *Meis2* expression depends on target-derived signals (**C, D**). (**C**) Representative images of *Meis2* mRNA expression (blue or pseudo-colored in red) and islet1 (green) in DRGs of Hamburger–Hamilton stage (HH) 36 chick embryos on the ablated and contralateral sides. Box plots showing the number of Islet1⁺/Meis2⁺ DRG neurons per section at stage HH36 following limb bud ablation. For Islet1-positive neurons, the contralateral side was considered as 100% of neurons per section. For Meis2-positive neurons, values represent the percentage of Meis2⁺ over Islet1⁺ neurons. (**D**) Representative images of *Meis2* mRNA expression (blue or pseudo-colored in red) and islet1 (green) in DRGs of HH29 chick embryos on the ablated and contralateral sides. Box plots showing the quantification of Islet1⁺/Meis2⁺ neurons number per section at stage HH29 on the contralateral and ablated sides. Arrowheads point

*Figure 1 continued on next page*

*Figure 1 continued*

at remaining Meis2-positive VL neurons. Dashed lines encircle the DRGs. **p≤0.005; ***p≤0.0005; ns = not significant following Student's *t*-test. n = 3 chick embryos. Scale bar = 100 μm. Altered touch perception in *Meis2* mutant mice (**E–H**). (**E**) Box plots showing the responses following application of Von Frey filaments of different forces. *Isl1⁺/Cre::Meis2LoxP/LoxP* mice exhibited a significantly reduced sensitivity to the 0.16, 0.4, and 0.6 g Von Frey filaments but not to higher forces filaments compared to WT and *Isl1⁺/Cre* littermates. * p≤0.05; ** p≤0.005; *** p≤0.001 following Kruskal–Wallis statistical analysis. (**F**) Box plots showing the dynamic touch responses when the hind paw palms of individual mice were stroked with a tapered cotton swab. Analysis showed that *Isl1⁺/Cre::Meis2LoxP/LoxP* mice were less responsive to the stimulus than WT and *Isl1⁺/Cre* littermates. *** p≤0.0001 following a one-way ANOVA statistical analysis. (**G**) Box plots indicating that the latency to the first signs of aversive behavior in the hot plate test is similar in all groups of mice. WT, n = 19; *Isl1⁺/Cre*, n = 16; *Isl1⁺/Cre::Meis2LoxP/LoxP*, n = 9. (**H**) Box plots showing the number of bouts when a sticky paper tape was applied on the back skin of mice. Analysis indicated a significant decrease in the number of bouts in *Isl1⁺/Cre::Meis2LoxP/LoxP* mice compared to WT and *Isl1⁺/Cre* littermates. * p≤0.05 following a one-way ANOVA statistical analysis.

The online version of this article includes the following source data and figure supplement(s) for figure 1:

**Source data 1.** *Isl1⁺/Cre::Meis2LoxP/LoxP* adult mice exhibit normal locomotion.

**Figure supplement 1.** *Meis2* mRNA expression in low-threshold mechanoreceptor (LTMR) neurons of mouse dorsal root ganglia (DRG).

**Figure supplement 2.** *Meis2* is expressed in a subset of chick ventrolateral dorsal root ganglia (DRG) sensory neurons during embryogenesis.

**Figure supplement 3.** Mice with a conditional deletion of *Meis2* gene in neural crest derivatives (Wnt1Cre) exhibited cleft palate and died at birth.

neurons were lost on the ablated side compared to the contralateral side (*Figure 1C*). This is consistent with a 30% loss of all sensory DRG neurons represented by the pan-neuronal marker Islet1, and the 50 and 65% loss of Ntrk2 and Ntrk3-positive VL-neurons, respectively (*Figure 1—figure supplement 2D*). The number of Ntrk2-positive DL neurons was not significantly affected. In HH27 embryos, while no significant loss of Islet1-positive neurons was detected following limb ablation, about 40% of Meis2-positive neurons were lost, and remaining *Meis2*-positive neurons expressed very low levels of *Meis2* mRNAs (*Figure 1D*).

These results indicate that target-derived signals are necessary for the maintenance but not the induction of Meis2 expression in sensory neurons.

## *Meis2* gene inactivation in postmitotic sensory neurons induces severe behavioral defects

We next asked whether *Meis2* inactivation would induce changes in LTMR structure and function. We generated a conditional mouse mutant strain for *Meis2* (*Meis2LoxP/LoxP*) in which the first coding exon for the homeodomain was flanked by *LoxP* sites (*Figure 1—figure supplement 3A*). To validate the use of our strain, we first crossed the *Meis2LoxP/LoxP* mice with the *Wnt1Cre* strain. This crossing efficiently inactivated *Meis2* in the neural crest, and *Wnt1Cre::Meis2LoxP/LoxP* newborn pups exhibited a cleft palate as previously reported in another conditional *Meis2* mouse strain (*Machon et al., 2015*; *Figure 1—figure supplement 3B*). They were, however, not viable, precluding functional and anatomical analyses at adult stages. To bypass this neural crest phenotype and more specifically address *Meis2* function in postmitotic neurons, we crossed the *Meis2LoxP/LoxP* mice with the Isl1Cre/+ strain and focused our analysis on the *Isl1Cre/+::Meis2LoxP/LoxP* strain. Mutant pups were viable, appeared healthy, and displayed a normal palate, allowing sensory behavior investigations.

We monitored tactile-evoked behaviors in adult WT, *Isl1Cre/+* and *Isl1Cre/+::Meis2LoxP/LoxP* mice using stimuli applied to both glabrous and hairy skin. We used Von Frey filaments to apply a series of low forces ranging from 0.008 to 1.4 g to the hind paw and found the frequency of withdrawal responses to be significantly decreased in *Isl1⁺/Cre::Meis2LoxP/LoxP* mice compared to control WT and *Isl1⁺/Cre* mice between 0.16 and 0.6 g (*Figure 1E*), indicating that mutant mice are less responsive to light touch. No differences were observed between *WT* and *Isl1⁺/Cre* mice for any of the stimuli applied. Behaviors evoked from stimulation of the glabrous skin were next assessed using the 'cotton swab' dynamic touch assay (*Bourane et al., 2015*). Here, responses were significantly decreased in Isl1⁺/Cre::Meis2LoxP/LoxP mice compared to control WT and Isl1⁺/Cre littermates (*Figure 1F*). We also used the hot plate assay to assess noxious heat-evoked behaviors and found no difference in response latencies between WT, *Isl1Cre/+* and *Isl1Cre/+::Meis2LoxP/LoxP* mice (*Figure 1G*). Finally, we compared the sensitivity of mice to stimuli applied to the hairy skin using the sticky tape test. Placing sticky tape on the back skin evoked attempts to remove the stimulus in a defined time window, and we found that such bouts of behavior were significantly reduced in *Isl1Cre/+::Meis2LoxP/LoxP* mice compared to WT and *Isl1Cre/+* control mice

(*Figure 1H*). Finally, although *Meis2* and *Isl1* are both expressed by spinal motor neurons and proprioceptors (*Catela et al., 2016*; *Dasen et al., 2005*; *Ericson et al., 1992*), we did not observe obvious motor deficits in *Isl1^{Cre/+}::Meis2^{LoxP/LoxP}* mice. Thus, in a catwalk analysis we found no differences in any of the gait parameters measured between WT and mutant mice (*Figure 1—source data 1*).

Overall, these behavioral analyses indicate that *Meis2* gene inactivation specifically affects light touch sensation both in the glabrous and the hairy skin. The impaired behavioral response to light touch in *Meis2* mutant suggests that *Meis2* gene activity is necessary for the anatomical and functional maturation of LTMRs.

## *Meis2* is dispensable for LTMR specification and survival

To investigate whether *Meis2* gene inactivation interfered with LTMR survival during embryonic development, we performed histological analysis of the *Isl1^{Cre/+}::Meis2^{LoxP/LoxP}* and *Wnt1^{Cre}::Meis2^{LoxP/LoxP}* strains (*Figure 2*). There was no difference in the size of the DRGs between E16.5 WT and *Isl1^{Cre/+}::Meis-2^{LoxP/LoxP}* embryos as well as in the number of Ntrk2 and Ntrk3-positive neurons (*Figure 2A*), suggesting no cell loss. In E18.5 embryonic DRGs, the number of LTMR and proprioceptors identified as positive for Ntrk2, Ntrk3, Ret, and Maf was unchanged following *Meis2* inactivation (*Figure 2C*). Consistent with the lack of *Meis2* expression in nociceptors, the number of Ntrk1-positive neurons was also unaffected (*Figure 2B*). Finally, quantification of DRG neuron populations at P0 in *Wnt1^{Cre}::Meis2^{LoxP/LoxP}* mice showed similar results with no differences in the number of Ntrk2 and Ntrk3-positive neurons (*Figure 2C*). At this stage, phospho-Creb (pCreb) expression in Ntrk2 and Ntrk3-positive neurons was similar in WT and mutants (*Figure 2—figure supplement 1*), suggesting that Ntrk signaling is not affected. Altogether, these results show that *Meis2* is dispensable for LTMR and proprioceptor survival and specification during embryogenesis.

## *Meis2* is necessary for normal end-organ innervation

To better understand the molecular changes underlying tactile defects in Meis2 mutant mice, we performed RNAseq analysis on DRGs dissected from WT, Isl1^{Cre/+}, and Isl1^{Cre/+}::Meis2^{LoxP/LoxP} E18.5 embryos.

For all analyses, consistent with the changes measured for *Meis2* and *Isl1* genes (*Figure 3—figure supplement 1A and B*), only DEGs with a minimal fold change of 20% and a p-value <0.05 were considered. Analyses of the dataset (n = 3; p<0.05; *Figure 3*, *Figure 3—figure supplement 1*, *Figure 3—source data 1*) identified 43 differentially expressed genes (DEGs) in the WT vs *Isl1^{+/Cre}::Meis2^{LoxP/LoxP}* comparison, 107 DEGs in the *Isl1^{+/Cre}* vs *Isl1^{+/Cre}::Meis2^{LoxP/LoxP}* comparison, and 109 DEGs in the WT vs *Isl1^{+/Cre}* comparison. Among them, only 10 DEGs were found in both WT vs *Isl1^{+/Cre}::Meis2^{LoxP/LoxP}* and *Isl1^{+/Cre}* vs *Isl1^{+/Cre}::Meis2^{LoxP/LoxP}* comparisons (*Figure 3A*). Half of them were down- or upregulated (*Figure 3—figure supplement 1C*), and eight were found to be expressed in sensory neurons expressing Meis2 (*Figure 3—figure supplement 1D*). These include three ncRNA (A230077H06Rik, Gm20163, Gm42418), the Adhesion G Protein-Coupled Receptor G3 (*Adgrg3*, also known as GPR97), the Cellular Repressor of E1A Stimulated Genes 2 (*Creg2*), predicted to be located in Golgi apparatus and endoplasmic reticulum, the Tubulin Alpha 8 (*Tuba8*) mutated in Polymicrogyria, a developmental malformation of the cortex (*Abdollahi et al., 2009*), the Hes Family BHLH Transcription Factor 5 (*Hes5*) activated downstream of the Notch pathway and largely involved in neuronal differentiation, the mitochondrial ribosomal protein s28 (*Mrps28*) whose mutation severely impairs the development of the nervous system (*Pulman et al., 2019*), the Phospholipase C Delta 1 (*Plcd1*) important for neuronal development and function of mature neurons, and the Pyridoxamine 5'-Phosphate Oxidase (Pnpo) involved in the synthesis of vitamin B6 and whose mutation causes a form of neonatal epileptic encephalopathy and motor neuron disease. Gene Ontology (GO) analysis for the 43 DEGs in the WT vs *Isl1^{+/Cre}::Meis2^{LoxP/LoxP}* comparison and for the 107 DEGs in the *Isl1^{+/Cre}* vs *Isl1^{+/Cre}::Meis2^{LoxP/LoxP}* comparison revealed significant relevant hits with many terms associated with neuronal projections and functions (*Figure 3B and C*, *Figure 3—source data 2*, *Figure 3—figure supplements 2 and 3*). These include subsets for the GO term associated with synapse, dendrite, and axons and more specifically with GABAergic synapses, dendritic shaft, or postsynaptic membrane. None of these GO terms were significantly enriched in the WT vs *Isl1^{+/Cre}* comparison (*Figure 3B*, *Figure 3—figure supplements 2 and 3*) which overall showed lower enrichment scores and p-values than in the two other datasets. It is important to note that many of the genes associated with neuron projection or

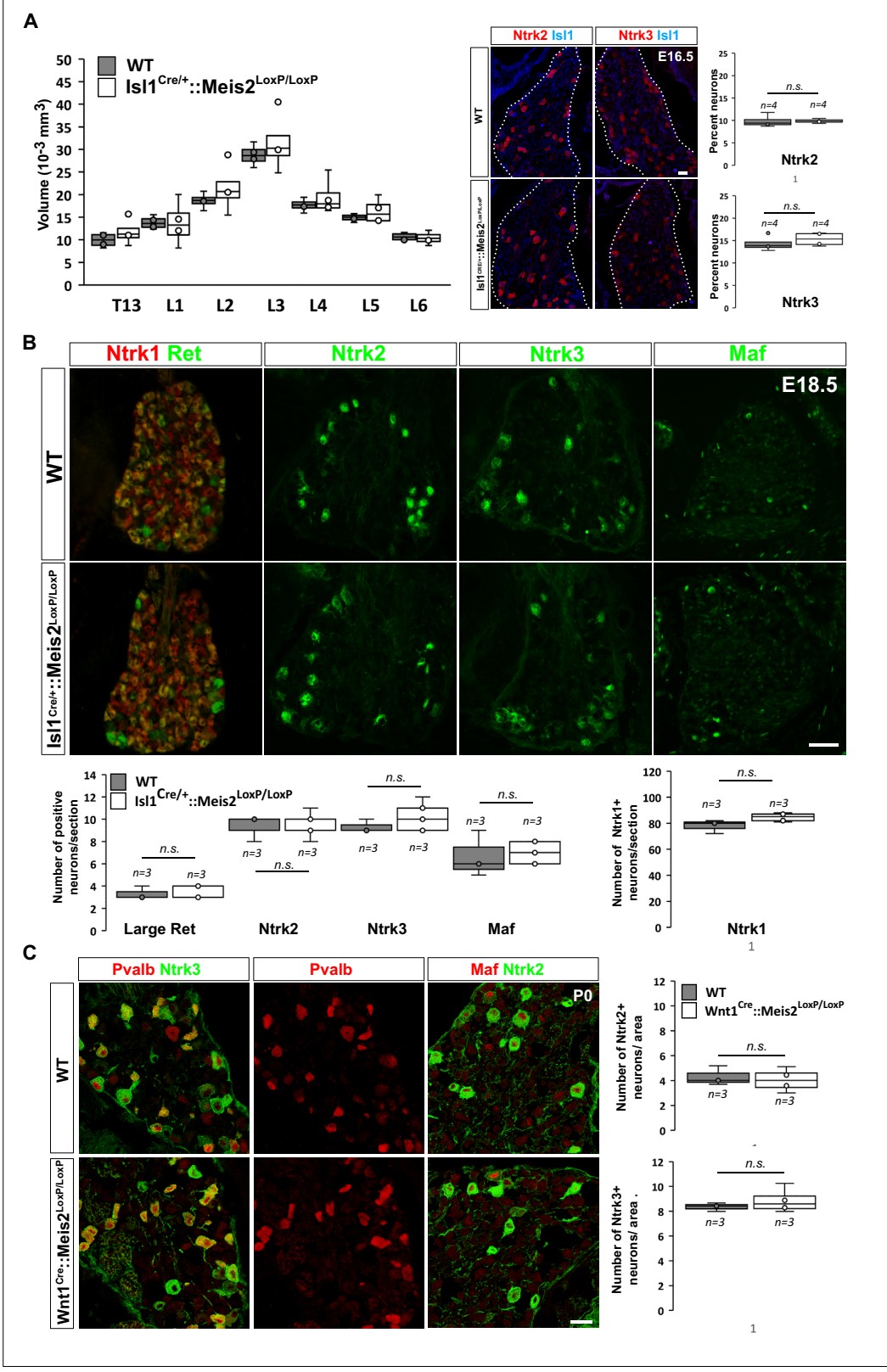

**Figure 2.** *Meis2* is dispensable for low-threshold mechanoreceptor (LTMR) neuron survival and specification. (**A**) Box plots showing that the dorsal root ganglia (DRG) volumes along the rostrocaudal axis are similar in embryonic day (E) 16.5 WT and *Isl1^Cre/+^::Meis2^LoxP/LoxP^* embryos. IF for Ntrk2 or Ntrk3 (red) and Islet1 (blue) and box plots analysis indicating that the percentage of Ntrk2+ and Ntrk3+ neurons is not affected in E16.5 *Isl1^Cre/+^::Meis2^LoxP/*

*Figure 2 continued*

*LoxP*. Dashed lines encircle the DRGs. n = 4; n.s. = not significant. Scale bar = 20 μm. (**B**) Representative images showing IF for Ntrk1, Ret, Ntrk2, Ntrk3, and Maf in E18.5 WT and *Isl1^Cre/+^::Meis2^LoxP/LoxP^* DRGs. Box plots showing that the number of Ret^+^, Ntrk2^+^, Ntrk3^+^, and Maf^+^ LTMR neurons and of Ntrk1^+^ nociceptive neurons are similar in E18.5 WT and *Isl1^Cre/+^::Meis2^LoxP/LoxP^* DRGs. n = 3; n.s. = not significant. Scale bar = 100 μm. (**C**) Representative images showing IF for Ntrk2 or Ntrk3 (green) with Pvalb or Maf (red) in P0 WT and Wnt1^Cre^::Meis2^LoxP/LoxP^ DRGs. Box plots showing that the number of Ntrk2^+^ and Ntrk3^+^ neurons is unchanged in P0 WT and *Wnt1^Cre^::Meis2^LoxP/LoxP^* DRGs. n = 3, n.s. = not significant. Scale bar = 20 μm.

The online version of this article includes the following figure supplement(s) for figure 2:

**Figure supplement 1.** *Meis2* gene inactivation does not affect phospho-Creb expression.

synapse that were present in either WT vs *Isl1^+/Cre^::Meis2^LoxP/LoxP^* dataset or *Isl1^+/Cre^* vs *Isl1^+/Cre^::Meis2^LoxP/LoxP^* dataset, such as *Oprd1*, *Calb2*, *Whrn*, *Lrp2*, *Lypd6*, *Grid1*, and *Rps21*, failed to enter the list of the 10 best DEGs either because their fold changes were below the cutoff or their p-values were close to but higher than 0.05. Interestingly, a significant association with the GO term Cadherin in the *Isl1^+/Cre^* vs *Isl1^+/Cre^::Meis2^LoxP/LoxP^* comparison points at the protocadherin family in which several members were downregulated (*Figure 3C*). Finally, comparing these genes to single-cell RNAseq (scRNAseq) analysis in adult DRG neurons (*Usoskin et al., 2015*) showed that most of them are expressed by Meis2-expressing DRG sensory neuron subtypes (*Figure 3—figure supplement 4*). These molecular analyses strongly support the role of *Meis2* in regulating embryonic target-field innervation. We thus investigated this hypothesis, and in P0 *Wnt1^Cre^::Meis2^LoxP/LoxP^*, Nefh staining in the hind paws showed strong innervation deficits as reflected by a paucity of neurofilament-positive myelinated branches in both the glabrous and hairy skin (*Figure 3E*). In WT newborn mice, numerous Nefh^+^ sensory fibers surround all dermal papillae of the hairy skin and footpad of the glabrous skin, whereas in *Wnt1^Cre^::Meis2^LoxP/LoxP^* littermates, very few Nefh^+^ sensory fibers are present and they poorly innervate the dermal papillae and footpads.

## *Meis2* gene is necessary for SA-LTMR morphology and function only in the glabrous skin

LTMRs form specialized sensory endings in a variety of end organs specialized to shape the mechanoreceptor properties. We used the *Isl1^+/Cre^::Meis2^LoxP/LoxP^* mice to assess the effects of late loss of *Meis2* on LTMR structure and function and investigate whether postmitotic *Meis2* inactivation impacts terminal morphologies and physiological properties of LTMRs. We made recordings from single mechanoreceptors and probed their responses to defined mechanical stimuli in adult WT, *Isl1^+/Cre^*, and *Isl1^+/Cre^::Meis2^LoxP/LoxP^* mice using ex vivo skin nerve preparations as previously described (*Schwaller et al., 2021*; *Walcher et al., 2018*; *Wetzel et al., 2007*).

We recorded single myelinated afferents in the saphenous nerve which innervates the hairy skin of the foot or from the tibial nerve that innervates the glabrous skin of the foot (*Schwaller et al., 2021*; *Walcher et al., 2018*). In control nerves, all the single units (n = 78) with conduction velocities in the Aβ-fiber range (>10 m s^-1^) could be easily classified as either rapidly adapting or slowly adapting mechanoreceptors (RA-LTMR or SA-LTMRs, respectively) using a set of standard quantitative mechanical stimuli. However, in the *Isl1^+/Cre^::Meis2^LoxP/LoxP^* mice about 10 and 18% of Aβ fibers in the hairy and glabrous skin respectively could not be reliably activated by any of the quantitative mechanical stimuli used. Sensory neurons that could not be activated by our standard array of mechanical stimuli but could still be activated by rapid manual application of force with a glass rod were classified as so-called 'tap' units (*Figure 4A*). Such 'tap' units have been found in several mice with deficits in sensory mechano-transduction (*Wetzel et al., 2007*; *Ranade et al., 2014*). We made recordings from SA-LTMRs from both glabrous and hairy skin, but decided to pool the data as there was an insufficient sample size from either skin area alone. We reasoned that electrophysiological recordings would pick up primarily receptors that had successfully innervated Merkel cells and miss those fibers that had failed to innervate end organs and would likely not be activated by mechanical stimuli. In this mixed sample of SA-LTMRs, the mean vibration threshold was significantly elevated in *Isl1^Cre/+^::Meis2^LoxP/LoxP^* mice, but it was clear that many fibers in this sample had mechanical thresholds similar to those in the wild type (*Figure 4B*). The response of the same SA-LTMRs to a 25 Hz sinusoidal stimulus was unchanged in *Isl1^Cre/+^::Meis2^LoxP/LoxP^* mice compared to controls (*Figure 4B*). The response of these

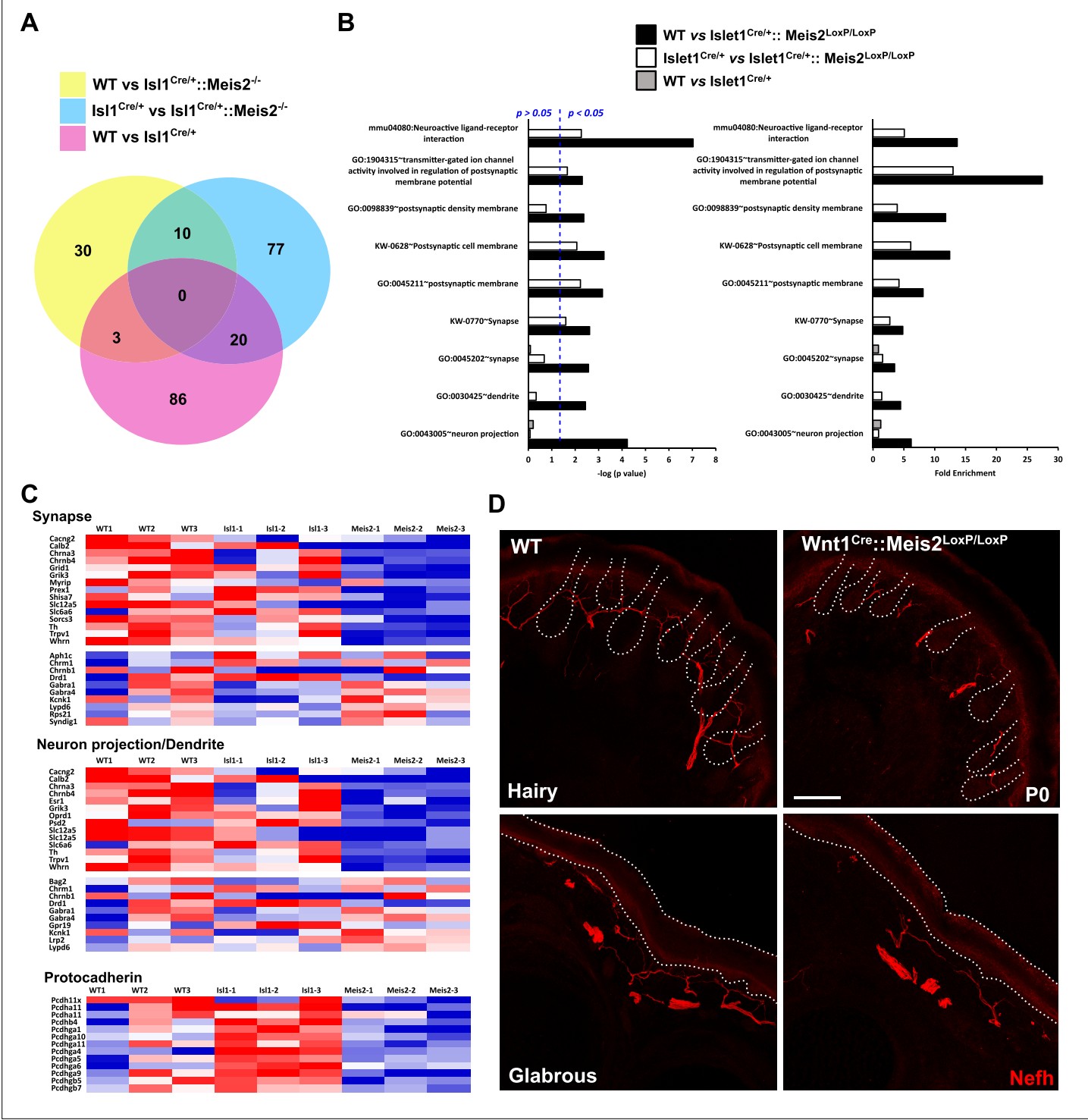

**Figure 3.** *Meis2* inactivation dysregulates genes linked to neuronal projections and synaptogenesis. (**A**) Venn diagram comparing the number of differentially expressed genes (DEGs) between each genotype (n = 3; p<0.05). This comparison identified 10 DEGs that were differentially expressed compared to both control genotypes (WT or *Isl1$^{+/Cre}$* embryos), and a total of 140 genes that were differentially expressed in *Meis2* mutant compared to either WT or *Isl1$^{+/Cre}$* embryos. (**B**) Gene Ontology (GO) analysis for the three paired-analysis (WT vs *Isl1$^{+/Cre}$::Meis2$^{LoxP/LoxP}$*; *Isl1$^{+/Cre}$* vs *Isl1$^{+/Cre}$::Meis2$^{LoxP/LoxP}$*, and WT vs *Isl1$^{+/Cre}$*) datasets using DAVID and the full RNAseq gene list as background. Graphs show the comparison of the fold enrichment and the -log10(p value) of selected significant (p<0.05) GO or KEGG_PATHWAY terms associated to synapse and neuron projections whatever the number of genes. Blue dotted line indicates a p-value of 0.05. Note that following DAVID analysis GO terms associated to synapse and neuron projections were overrepresented in the WT vs *Isl1$^{+/Cre}$::Meis2$^{LoxP/LoxP}$* and the *Isl1$^{+/Cre}$* vs *Isl1$^{+/Cre}$::Meis2$^{LoxP/LoxP}$* datasets compared to the WT vs *Isl1$^{+/Cre}$* dataset. (**C**)

*Figure 3 continued on next page*

*Figure 3 continued*

Heat maps showing the DEGs related to the GO terms synapse, neuron projection including dendrite, and protocadherin. (**D**) Representative images showing a strong overall deficit of Nefh⁺ (red) sensory projections innervating the dermal papillae in the hairy and the foot pads in the glabrous skin of P0 *Wnt1^{Cre}::Meis2^{LoxP/LoxP}* neonates forepaw compared to WT littermates. Dashed lines delineate the hair follicle and the epidermis. Scale bar = 50 µm.

The online version of this article includes the following source data and figure supplement(s) for figure 3:

**Source data 1.** Table showing the results of the bulk RNAseq analysis.

**Source data 2.** Table showing the results of the GO terms analysis performed with AVID.

**Figure supplement 1.** Best dysregulated genes in Meis2 mutant adult DRG.

**Figure supplement 2.** Best DAVID GO terms for each dataset in the RNAseq paired analysis.

**Figure supplement 3.** Best DAVID terms other than GO for each dataset in the RNAseq paired analysis.

**Figure supplement 4.** Comparison of differentially expressed genes (DEGs) in Meis2 mutant embryonic dorsal root ganglia (DRG) neurons with their expression in the different adult DRG sensory neuron subtypes.

fibers to ramp stimuli of increasing velocities or to increasing amplitudes of ramp and hold stimuli was also not significantly different in mutant mice compared to controls (*Figure 4C*).

In both the glabrous and the hairy skin, Merkel cells are innervated by slowly adapting mechanoreceptor type I (SAI-LTMR) neurons responding to both static skin indentation and moving stimuli such as vibration. In the glabrous skin, Merkel cells form clusters in the basal layer of the epidermis, and in the hairy skin, similar clusters of Merkel cells called touch domes are located at the bulge region of guard hairs. Histological analysis indicated that in the forepaw glabrous skin of *Isl1^{+/Cre}::Meis2^{LoxP/LoxP}* adult mice, the number of Merkel cells contacted by Nefh-positive fibers was strongly decreased compared to *Isl1^{+/Cre}* (*Figure 4D*). However, in contrast to the glabrous skin, Merkel cell innervation by Nefh-positive fibers appeared largely unaffected in the hairy skin of *Isl1^{Cre/+}::Meis2^{LoxP/LoxP}* mice (*Figure 4E*). Whole-mount analysis of CK8-positive Merkel cells in the hairy back skin of E18.5 embryos showed that the overall number of touch domes and Merkel cells per touch dome was unchanged in mutant animals compared to WT (*Figure 4F*).

Altogether, these data indicate that *Meis2* is necessary for Merkel cell innervation in the glabrous, but not in the hairy skin. In addition, electrophysiological recordings indicate that among SA-LTMRs, there was a light loss of sensitivity that could be associated with poor innervation of Merkel cells in the glabrous skin.

## *Meis2* is necessary for RA-LTMR structure and function

In the glabrous skin, Meissner corpuscles are located in the dermal papillae and are innervated by rapidly adapting type LTMR (RA-LTMR) that detect small-amplitude skin vibrations <80 Hz.

Histological analysis of the glabrous skin showed that Nefh-positive innervation of the Meissner corpuscles was strongly disorganized (*Figure 5A*, *Figure 5—video 1; 2*) with decreased complexity of the Nefh⁺ fibers within the corpuscle as shown by quantification of the average number of time that fibers cross the midline of the terminal structure. However, recordings from RA-LTMRs innervating these structures in *Isl1^{Cre/+}::Meis2^{LoxP/LoxP}* animals showed largely normal physiological properties (*Figure 5B*). Thus RA-LTMRs recorded from *Isl1^{Cre/+}::Meis2^{LoxP/LoxP}* displayed normal vibration sensitivity in terms of absolute threshold and their ability to follow 25 sinusoids. There was a tendency for RA-LTMRs in *Isl1^{Cre/+}::Meis2^{LoxP/LoxP}* mutant mice to fire fewer action potentials to sinusoids and the ramp phase of a series 2 s duration ramp and hold stimuli, but these differences were not statistically significant (*Figure 5B*).

In the hairy skin, RA-LTMRs form longitudinal lanceolate endings parallel to the hair shaft of guard and awl/auchene hairs and respond to hair deflection only during hair movement, but not during maintained displacement (*Lechner and Lewin, 2013*). Similar to Meissner corpuscles, they are tuned to frequencies between 10 and 50 Hz (*Schwaller et al., 2021*). Whole-mount analysis of Nefh-positive fibers in the adult back skin showed an overall decrease in the innervation density of hairs in *Isl1^{Cre/+}::Meis2^{LoxP/LoxP}* animals compared to *Isl1^{Cre/+}* (*Figure 6A*). Our analysis revealed significant decreases in both the number of plexus branch points and the number of innervated hair follicles (*Figure 6B*).

Consistent with the hypo-innervation of hair follicles in *Isl1^{Cre/+}::Meis2^{LoxP/LoxP}*, we observed robust deficits in the mechanosensitivity of RA-LTMRs in the hairy skin (*Figure 6C* ). Thus, we needed sinusoids of significantly larger amplitudes to evoke the first (threshold) spike in RA-LTMRs. We therefore

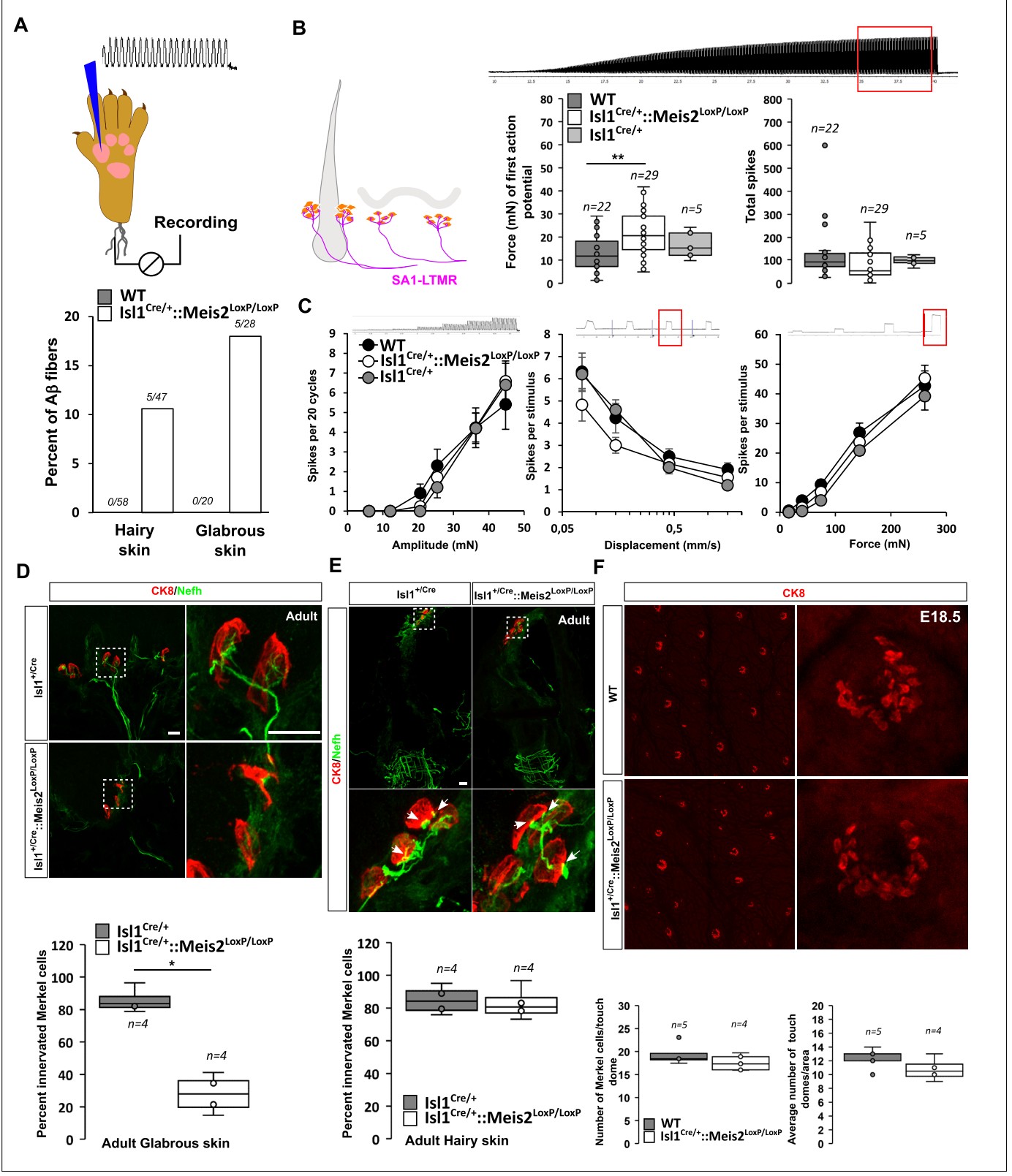

**Figure 4.** *Meis2* gene inactivation compromised Merkel cell innervation in the glabrous skin and increased slowly adapting mechanoreceptor (SAM) vibration threshold. (**A**) Graph showing the percentage of tap units among all recorded Aβ fibers in the nerve skin preparation both in the hairy and glabrous skins. The number of tap units over the number of recorded fibers is indicated. Note that Tap units are only present in both the hairy and glabrous skin of adult *Isl1+/Cre::Meis2LoxP/LoxP* mice but not in WT littermates. (**B**) In the hairy and glabrous skins, SAMs in *Isl1+/Cre::Meis2LoxP/LoxP* mice (n

*Figure 4 continued on next page*

*Figure 4 continued*

= 22 from six mice) had significantly increased vibration threshold compared to WT mice (n = 29 from six mice), but normal firing activity to a 25 Hz vibration. Trace shows the stimulation applied to the skin, and red squares indicate the time frame during which the number of spikes was calculated. (**C**) SAM responses to a ramp of 50 Hz vibration with increasing amplitude are similar in WT, $Isl1^{+/Cre}$, and $Isl1^{+/Cre}::Meis2^{LoxP/LoxP}$ mice. SAM responses to ramp stimuli and their static force responses were also identical in the different genotypes. Fibers from WT and $Isl1^{+/Cre}$ mice (n = 5) displayed similar responses. * p≤0.05; ** p≤0.005. Traces show the applied stimulus and red squares the time frame during which the parameters below were measured. (**D**) Confocal images of Nefh$^+$ innervation (green) of CK8$^+$ Merkel cells (red) in the forepaw glabrous skin of $Isl1^{Cre/+}$and $Isl1^{Cre/+}::Meis2^{LoxP/LoxP}$ adult mice. Dotted white squares indicate the close-up of CK8$^+$ Merkel cells. Note the lack of Nefh$^+$ fibers innervating Merkel cells in mutant mice. White arrows point at contact between NF200$^+$ fibers and CK8$^+$ Merkel cells. Scale bar = 10 μm. The box plot indicates the percentage of Merkel cells in contact with Nefh$^+$ fibers. n = 4. * p≤0.05 in Mann–Whitney test. (**E**) Confocal images of Nefh$^+$ innervation (green) and CK8$^+$ Merkel cells (red) of guard hairs in the hairy back skin of $Isl1^{Cre/+}$and $Isl1^{Cre/+}::Meis2^{LoxP/LoxP}$ adult mice. Dotted white squares indicate the close-up of CK8$^+$ Merkel cells with apparently normal Nefh$^+$ innervation. White arrows point at contacts between Nefh$^+$ fibers and CK8$^+$ Merkel cells. Scale bar = 10 μm. The box plot indicates the percentage of Merkel cells contacted by Nefh$^+$ fibers. n = 4. (**F**) Representative images of whole-mount staining for CK8 in the hairy back skin of WT and $Isl1^{Cre/+}::Meis2^{LoxP/LoxP}$ E18.5 embryos showing no difference in the number of touch dome between genotypes. Box plots show the number of touch domes per surface area and the number of Merkel cells per touch dome. No significant differences were found between both genotypes in Mann–Whitney test. n = 5 (WT) and 4 ($Isl1^{+/Cre}::Meis2^{LoxP/LoxP}$).

measured the total number of spikes evoked by a sinusoid stimulus (25 Hz) of gradually increasing amplitude. Again, RA-LTMRs fired considerably less in $Isl1^{+/Cre}::Meis2^{LoxP/LoxP}$ mutant than in control mice. This finding was confirmed using a series of vibration steps of increasing amplitudes again demonstrating decreased firing in response to 25 Hz vibration stimuli (**Figure 6C**). Thus, the functional deficits in RA-LTMRs correlate well with the defects in LTMR cutaneous projections we observed in $Isl1^{+/Cre}::Meis2^{LoxP/LoxP}$ mutant mice.

Finally, D-hair mechanoreceptors or Aδ−LTMRs are the most sensitive skin mechanoreceptors with very large receptive fields (**Li et al., 2011**; **Walcher et al., 2018**; **Shin et al., 2003**). They form lanceolate endings, are thinly myelinated, and are activated by movement of the smaller zigzag hairs (**Lechner and Lewin, 2013**). Consistent with the lack of Meis2 expression in this population reported by scRNAseq databases, Aδ fibers D-hair in the hairy skin showed similar vibration responses in WT and $Isl1^{+/Cre}::Meis2^{LoxP/LoxP}$ mice (**Figure 5—figure supplement 1**).

## Discussion

The function of the Meis family of TFs in postmitotic neurons has only been marginally addressed (**Bouilloux et al., 2016**; **Jakovcevski et al., 2015**; **Agoston et al., 2014**). Here, we showed that *Meis2* is selectively expressed by subpopulations of early postmitotic cutaneous LTMR and proprioceptive neurons during the development of both mouse and chick, highlighting the conserved Meis2 expression across vertebrate species in those neurons. Our results on Meis2 expression are in agreement with previous combined scRNAseq analysis and genetic tracing reporting Meis2 in proprioceptive neurons, Aβ field-LTMR, Aβ-SA1-LTMR, and Aβ-RA-LTMR, but not in C-LTMR, Aδ-LTMR, peptidergic, and non-peptidergic nociceptive neurons (**Sharma et al., 2020**; **Usoskin et al., 2015**; **Zheng et al., 2019**). We unambiguously demonstrate that *Meis2* differentially regulates target-field innervation and function of postmitotic LTMR neurons. *Meis2* inactivation in postmitotic sensory neurons modified their embryonic transcriptomic profile and differentially impaired adult LTMR projections and functions without affecting their survival and molecular subtype identity.

The morphological and functional phenotypes we report following specific *Meis2* gene inactivation in postmitotic sensory neurons are consistent with its expression pattern, and ultimately, both defective morphological and electrophysiological responses result in specifically impaired behavioral responses to light touch mechanical stimuli. In these mutants, the decreased innervation of Merkel cells in the glabrous skin and the decreased sensitivity in SA-LTMR electrophysiological responses to mechanical stimuli are consistent with *Meis2* being expressed by Aβ-SA1-LTMR neurons. Interestingly, *Meis2* gene inactivation compromises Merkel cell innervation and electrophysiological responses in the glabrous skin but not in touch domes of the hairy skin where innervation appeared unchanged. This difference supports previous work suggesting that the primary afferents innervating Merkel cells in the glabrous and the hairy skin maybe different (**Niu et al., 2014**; **Olson et al., 2016**). Whereas Merkel cells of the glabrous skin are exclusively contacted by large Ntrk3/Nefh-positive Aβ afferents, neonatal mouse touch domes receive innervation of two types of neuronal populations, a Ret/Ntrk1-positive one that

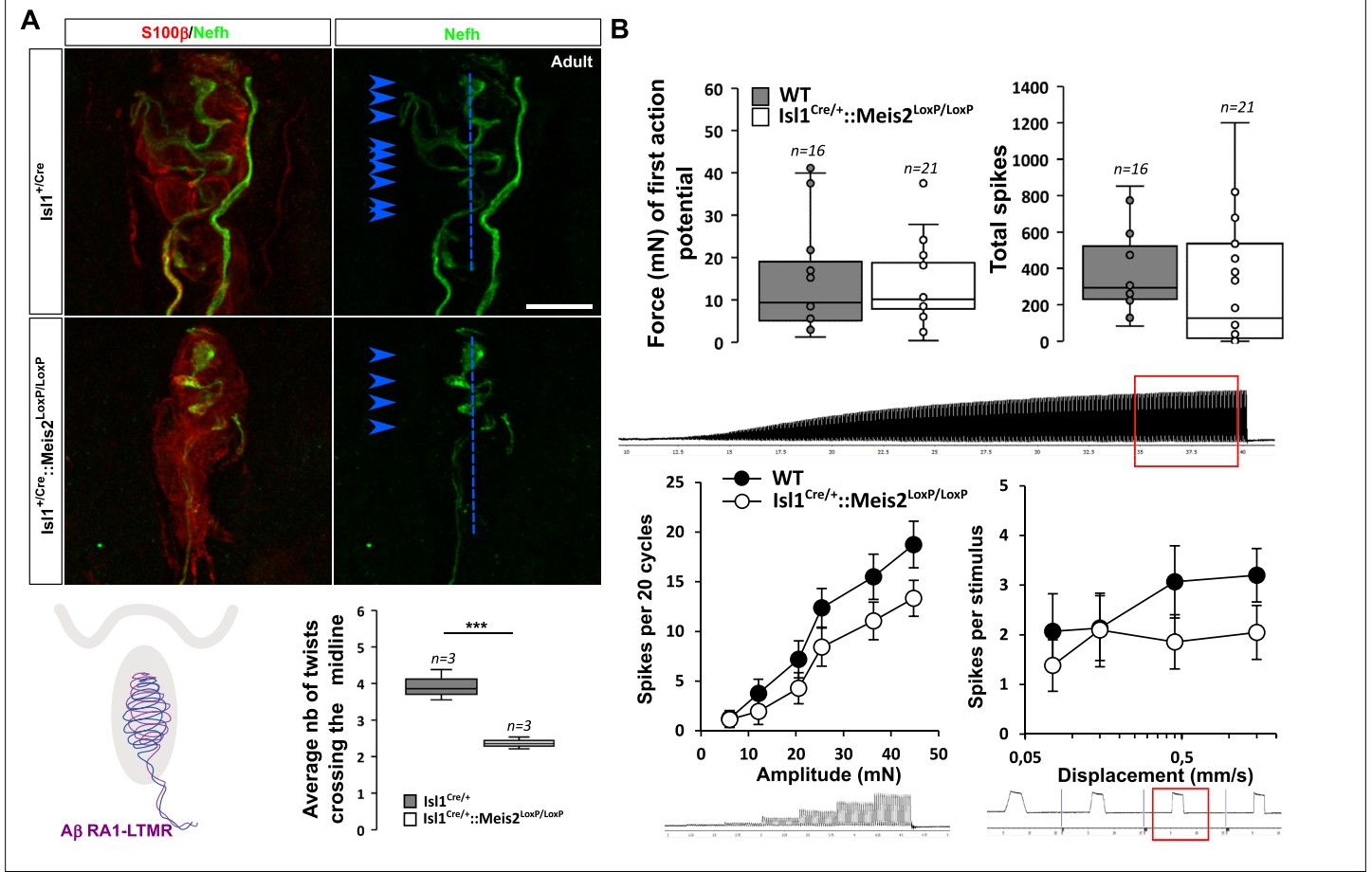

**Figure 5.** *Meis2* gene inactivation affects Meissner corpuscles morphology. (**A**) Representative images showing S100β⁺ Meissner corpuscles (red) and their innervation by Nefh⁺ fibers (green) in the glabrous skin of WT and *Isl1+/Cre::Meis2LoxP/LoxP* adult mice. Scale bar = 10 μm. The box plot shows the average number of times Nefh⁺ fibers cross the midline of the Meissner corpuscles. Dashed blue lines indicate the Meissner corpuscle midline. Blue arrowheads indicate sites where Nefh⁺ fibers cross this midline. (**B**) RAMs of the glabrous skin exhibited similar vibration threshold and firing activity to a 25 Hz vibration in WT (n = 16 from four mice) and *Isl1+/Cre::Meis2LoxP/LoxP* mice (n = 21 from six mice). Glabrous RAMs showed a nonsignificant decrease in firing activity to a ramp of 50 Hz vibration with increasing amplitude in *Isl1+/Cre::Meis2LoxP/LoxP* compared to WT littermates, but their response to ramp stimuli was similar in both genotypes. Traces indicate the type of stimulation and red squares the time frame during which the number of spikes was calculated. *** p≤0.001; Student's *t*-test.

The online version of this article includes the following video and figure supplement(s) for figure 5:

**Figure supplement 1.** Normal electrophysiological responses of D-hair mechanoreceptors following *Meis2* gene inactivation.

**Figure 5—video 1.** Meissner corpuscle in WT.

https://elifesciences.org/articles/89287/figures#fig5video1

**Figure 5—video 2.** Meissner corpuscles in Isl1+/Cre::Meis2LoxP/LoxP.

https://elifesciences.org/articles/89287/figures#fig5video2

depends on Ntrk1 for survival and innervation, and another Ntrk3/Nefh-positive that does not depend on Ntrk1 signaling during development (*Niu et al., 2014*). However, the functional significance of these different innervations is unknown. Denervation in rat also pointed to the differences between Merkel cells of the glabrous and the hairy skin. Following denervation, Merkel cells of the touch dome almost fully disappear, whereas in the footpad, Merkel cells developed normally (*Mills et al., 1989*).

Because touch domes innervation and Aδ fibers D-hair vibration responses were unaffected in *Isl1+/Cre::Meis2LoxP/LoxP* mice, we postulate that the innervation defects we observed in the hairy skin are supported by defects in lanceolate endings with RA-LTMR electrophysiological properties. However, the increased number of 'tap' units both in the hairy and glabrous skin is compatible with wider deficits also including Aβ-field LTMRs peripheral projections. Similarly, although the severely disorganized

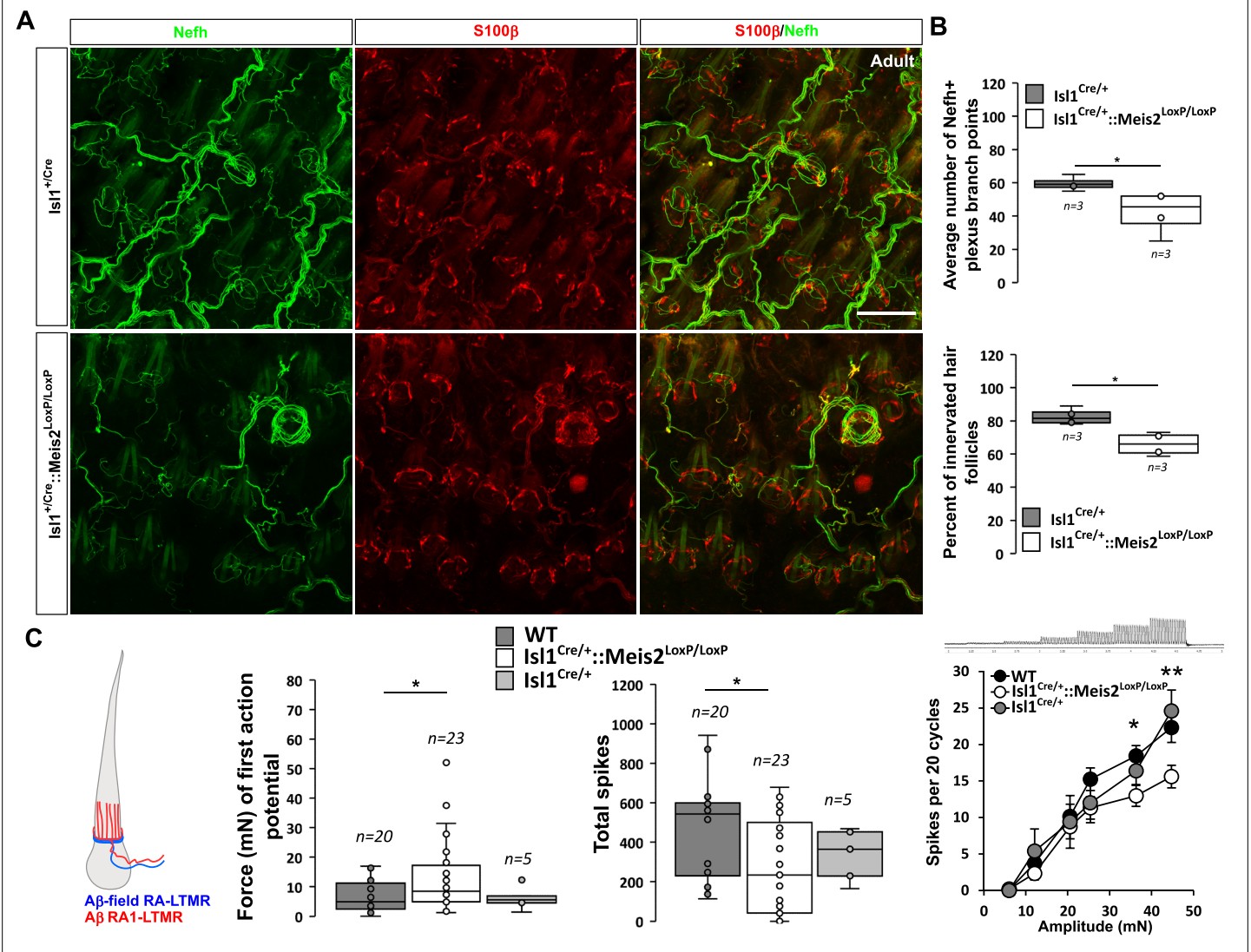

**Figure 6.** *Meis2* gene inactivation affects hair follicle innervation and RAM fibers electrophysiological responses in the hairy skin. (**A**) Representative images of whole-mount immunostaining for Nefh+ sensory projections (green) in the hairy skin of adult WT and *Isl1+/Cre::Meis2LoxP/LoxP* embryos counterstained with S100β (red) to highlight terminal Schwann cells decorating the periphery of hair follicles. Scale bar = 100 μm. (**B**) Box plots showing the quantification for the number of branch points in the innervation network and the number of innervated hair follicles. n = 3; *p≤0.05. (**C**) RAMs in the hairy skin of *Isl1Cre::Meis2LoxP/LoxP* mice (n = 24 from three mice) exhibited significantly increased vibration threshold and reduced firing activity to a 25 Hz vibration compared to WT mice (n = 20 from three mice). RAMs in the hairy skin of *Isl1+/Cre::Meis2LoxP/LoxP* mice also showed a reduced firing activity in response to a ramp of 50 Hz vibration with increasing amplitude compared to WT and *Isl1+/Cre* animals. Fibers recorded from *Isl1+/Cre* mice (n = 5) showed similar responses than those recorded from WT mice. * p≤0.05; **p≤0.005.

Meissner corpuscle architecture did not result in significant consequences on RAM fibers' electrophysiological responses in the glabrous skin, it is possible that the large increase in the number of 'tap units' within Aβ fibers of the glabrous skin represents Meissner corpuscles whose normal electrophysiological responses are abolished. Indeed, the electrophysiology methods used here can only identify sensory afferents that have a mechanosensitive receptive field. Primary afferents that have an axon in the skin but no mechanosensitvity can only be identified with a so-called electrical search protocol (*Wetzel et al., 2007*; *Ranade et al., 2014*) which was not used here. It is therefore quite likely that many primary afferents that failed to form endings would not be recorded in these experiments, for example, SA-LTMRs and RA-LTMRs that fail to innervate end organs (*Figures 4–6*). In agreement, challenging sensory responses in the glabrous skin with either Von Frey filament application or cotton swab stroking clearly showed a dramatic loss of mechanical sensitivity specifically within the range of

gentle touch neurons. Recent work reported that the Von Frey test performed within low forces and challenging light touch sensation could distinguish Merkel cells from Meissner corpuscles dysfunctions. Mice depleted of Merkel cells performed normally on this test while mice mutated at the *Ntrk2* locus with Meissner corpuscle innervation deficits were less sensitive in response to filament within the 0.02–0.6 g range (*Neubarth et al., 2020*). Thus, our result in the Von Frey test likely reflects aberrant functioning of the RAM LTMR-Meissner corpuscle complex. Finally, the unaltered D-hair fibers' electrophysiological responses and the normal noxious responses in the hot plate setting are consistent with the absence of *Meis2* expression in Aδ-LTMR, peptidergic, and non-peptidergic neurons. Surprisingly, although *Meis2* is expressed in proprioceptive neurons (*Usoskin et al., 2015*; *Zheng et al., 2019*; *Shin et al., 2020*), their function appeared not to be affected as seen by normal gait behavior in catwalk analysis. This is in agreement with studies in which HoxC8 inactivation, a classical Meis TF co-factor expressed by proprioceptive neurons from E11.5 to postnatal stages, affected neither proprioceptive neurons early molecular identity nor their survival (*Shin et al., 2020*). From our data, we could not conclude whether SA-LTMR electrophysiological responses are differentially affected in the glabrous vs hairy skin of *Meis2* mutant as suggested by histological analysis. Similarly, the decreased sensitivity of *Meis2* mutant mice in the cotton swab assay and the morphological defects of Meissner corpuscles evidenced in histological analysis do not correlate with RA-LTMR electrophysiological responses for which a tendency to decreased responses was, however, measured. The latter might result from an insufficient number of fibers recording, whereas the first may be due to pooling SA-LTMR from both the hairy and glabrous skin.

Understanding the transcriptional programs controlling each step in the generation of a given fully differentiated and specified neuron is an extensive research field in developmental neurobiology. Basic studies in model organisms led to a functional classification of TFs. Proneural TFs such as *Neurogenins* control the expression of generic pan-neuronal genes and are able to reprogram nearly any cell types into immature neurons (*Guillemot and Hassan, 2017*; *Baker and Brown, 2018*). Terminal selectors are TFs mastering the initiation and maintenance of terminal identity programs through direct regulation of neuron type-specific effector genes critical for neuronal identity and function such as genes involved in neurotransmitters synthesis and transport, ion channels, receptors, synaptic connectivity, or neuropeptide content (*Hobert, 2016*; *Hobert and Kratsios, 2019*). The proneural function of *Ngn1* and *Ngn2* genes in neural crest cells, the precursors of DRG sensory neurons, is well demonstrated (*Marmigère and Ernfors, 2007*; *Ma et al., 1999*), and several terminal selector genes shaping the different DRG sensory subpopulations have also been clearly identified including for cutaneous LTMRs (*Lallemend and Ernfors, 2012*; *Marmigère and Ernfors, 2007*; *Abdo et al., 2011*; *Olson et al., 2016*; *Wende et al., 2012*; *Yoshikawa et al., 2013*; *Lin et al., 1998*; *Inoue et al., 2002*). *Maf*, *Runx3*, *Shox2*, *ER81*, and *Pea3* are part of this regulatory transcriptional network regulating cutaneous LTMR neuron diversification through intermingled crossed activation and/or repression of subclasses-specific effector genes (*Marmigère and Carroll, 2014*).

In humans, at least 17 different mutations in the *Meis2* gene have been associated with neurodevelopmental delay (*Gangfuß et al., 2021*; *Giliberti et al., 2020*; *Shimojima et al., 2017*), emphasizing its essential function in neuronal differentiation. *Meis2* function in late differentiation of postmitotic peripheral sensory neurons adds to the wide actions of this TF in the developing and adult nervous system in the number of regions of the mouse nervous system. Its expression both in dividing neural progenitors, immature neurons, and discrete populations of mature neurons (*Jakovcevski et al., 2015*; *Toresson et al., 1999*; *Toresson et al., 2000*; *Chang and Parrilla, 2016*; *Frazer et al., 2017*; *Bumsted-O'Brien et al., 2007*; *Yan et al., 2020*; *Yang et al., 2021*; *Allen et al., 2007*) argues for diverse functions ranging from regulation of neuroblasts cell-cycle exit, to cell-fate decision, neurogenesis, neuronal specification, neurite outgrowth, synaptogenesis, and maintenance of mature neurons. In DRG LTMRs, *Meis2* fulfills some but not all of the criteria defining terminal selector genes. Its inactivation in neural crest cells does not affect sensory neuron generation nor pan-neuronal features, clearly excluding it from a proneural TF function. Although *Meis2* expression is continuously maintained in defined sensory neuron subtypes starting from early postmitotic neurons throughout life, its expression is not restricted to a unique neuronal identity and its early or late inactivation in either sensory neuron progenitors or postmitotic neurons does not influence neuronal subtypes identity nor survival as seen by the normal numbers of Ntrk2, Ntrk3, or Ret-positive neurons. However, our transcriptomic analyses strongly support that in LTMRs, Meis2 regulates other types of terminal effector

genes such as genes participating in neurotransmitter machinery specification and/or recognition, establishment, and maintenance of physical interactions between LTMRs and their peripheral targets. Surprisingly, our RNAseq analysis only revealed 10 DEGs that could be unambiguously attributed to Meis2 activity. All of these genes are expressed in adult LTMRs (*Usoskin et al., 2015*), suggesting that they exert specific functions in the maintenance of these sensory subclasses, and that their up- or downregulation might affect LTMRs maturation, but did not show any GO term enrichment. Separate GO analyses of our datasets, however, revealed alterations in pathways associated to synapse function and neuron projections. Therefore, from our results, we cannot exclude that dysregulation of those genes is secondary to the changed expression of one or more of the 10 above DEGs. Two GABA(A) receptor subunits (GABRA1 and GABRA4), the K-Cl cotransporter SLC12A5 associated to GABAergic neurotransmission, but also the glutamate receptor subunits GRID1 and GRIK3 are down- or upregulated in Meis2 mutants, questioning whether an imbalance gabaergic and glutamatergic transmission is responsible for Meis2 sensory phenotypes. Interestingly, Meis2 inactivation seems to interfere with the embryonic expression of many members of the protocadherin family, and the protocadherin γ cluster (Pcdhg) in particular has recently been highlighted as essential for building central and peripheral LTMRs innervation and synapses and establish proper peripheral target-field innervation and touch sensation (*Meltzer et al., 2023*). Finally, previous work on *Islet1* conditional deletion in DRG sensory neurons reports considerable changes in gene expression. Early homozygous *Islet1* deletion results in increased sensory neuron apoptosis and a loss of Ntrk1- and Ntrk2-positive neurons, whereas late deletion seems to only affect the nociceptive subpopulations (*Sun et al., 2008*). Whereas the distal projection defects we report in $Wnt^{Cre}::Meis2^{LoxP/LoxP}$ mutant can only be attributed to Meis2 inactivation, it is possible that among the DEGs we identified in $Isl1^{Cre/+}::Meis2^{LoxP/LoxP}$, some are epistatically regulated by the heterozygous *Islet1* deletion in addition to *Meis2* homozygous deletion. Such epistasis has been previously shown for Islet1 and the transcription factor Brn3a (*Sun et al., 2008*; *Dykes et al., 2011*).

In conclusion, this study reveals a novel function for the Meis2 transcription factor in selectively regulating target-field innervation of LTMR neurons. More broadly, it opens new perspectives to molecularly understand how Meis2 is linked to neuronal development. Together with studies on Meis2 function in the SVZ where it is necessary to maintain the neurogenic effect of Pax6 in neural progenitors and is later expressed in their mature progenies (*Agoston et al., 2014*), our results raise the possibility that this TF sets up a lineage-specific platform on which various specific co-factors in turn participate in additional and/or subsequent steps of the neuronal differentiation program.

## Materials and methods
### Animals
All procedures involving animals and their care were conducted according to European Parliament Directive 2010/63/EU and the September 22, 2010, Council of the Protection of Animals, and were approved by the French Ministry of Research (APAFIS#17869-2018112914501928v2, June 4, 2020).

### Mice strains
Wnt1$^{Cre}$ and Islet1$^{+/Cre}$ mice were previously described (*Yang et al., 2006*; *Lewis et al., 2013*). We previously reported the conditional mutant strain for Meis2 (Meis2$^{LoxP/LoxP}$) used in the present study (*Roussel et al., 2022*). To generate this strain, exon 8 of the *Meis2* gene was flanked by the LoxP recognition elements for the Cre recombinase at the ITL (Ingenious Targeting Laboratory, NY) using standard homologous recombination technology in mouse embryonic stem cells. FLP-FRT recombination was used to remove the neomycin selection cassette, and the Meis2$^{LoxP/LoxP}$ mutant mice were backcrossed for at least eight generations onto the C57BL/6 background before use. The primers used to genotype the different strains were Meis2 sense 5'-TGT TGG GAT CTG GTG ACT TG-3'; Meis2 antisense 5'-ACT TCA TGG GCT CCT CAC AG-3'; Cre sense 5'-TGC CAG GAT CAG GGT TAA AG-3'; Cre antisense 5'-GCT TGC ATG ATC TCC GGT AT-3'. Mice were kept in an animal facility, and gestational stages were determined according to the date of the vaginal plug.

For retro-tracing experiments, newborn pups were anesthetized on ice and CTB coupled to Cholera Toxin Subunit B conjugated with Alexa Fluor 488 (Thermo Fisher) was injected using a glass

micropipette in several points of the glabrous and hairy forepaw. Mice were sacrificed 7 d after injection and L4 to L6 DRGs were collected for analysis.

For behavioral assays, skin-nerve preparation, and electrophysiological recording, sex-matched 12-week-old mutant and WT littermates mice were used.

## Chick

Fertilized eggs were incubated at 37°C in a humidified incubator. For limb ablation experiments, eggs were opened on the third day of incubation (embryonic day 3, stage 17/18) (*Hamburger and Hamilton, 1992*) and the right hind limb bud was surgically removed as previously reported (*Oakley et al., 1995*). Eggs were closed with tape and further grown in the incubator for four (HH27) or seven (HH36) additional days before collection.

## Tissue preparation

Mouse and chick embryos were collected at different stages, fixed in 4% paraformaldehyde/PBS overnight at 4°C, and incubated overnight at 4°C for cryopreservation in increasing sucrose/PBS solutions (10–30% sucrose). After snap freezing in TissueTek, embryos were sectioned at 14 µm thickness and stored at –20°C until use.

## Cloning of mouse and chick Meis2 and probes preparation

For preparation of digoxigenin- and fluorescein-labeled probes, RNA from whole mouse or chick embryos was extracted using Absolutely RNA Nanoprep kit (Stratagene) following the manufacturer's instruction. Reverse transcription (RT) was carried out for 10 min at 65°C followed by 1 hr at 42°C and 15 min at 70°C in 20 µl reactions containing 0.5 mM dNTP each, 10 mM DTT, 0.5 µg oligod(T)15 (Promega), and 200 U of Super Script II RT (Gibco BRL Life Technologies). A 1206-bp-long and 1201-bp-long Meis2 fragments were amplified from mouse and gallus cDNA respectively using the following primers: mMeis2 forward: 5'-ATGGCGCAAAGGTACGATGAGCT-3'; mMeis2 reverse: 5'-TTACATAT AGTGCCACTGCCCATC-3'; gMeis2 forward: 5'-ATGGCGCAAAGGTACGATGAG-3'; gMeis2 reverse: 5'-TTACATGTAGTGCCATTGCCCAT-3'. PCRs were conducted in 50 µl reactions containing 10% RT product, 200 µM each dNTP, 10 pmol of each primer (MWG-Biotech AG), 3 mM MgCl$_2$, 6% DMSO, and 2.5 U of Herculase hotstart DNA polymerase (Stratagene). cDNA was denatured for 10 min at 98°C and amplified for 35 cycles in a three-step program as follows: 1 min denaturation at 98°C, 1 min annealing at annealing temperature and then 1.5 min polymerization at 72°C. PCR products were separated into 2% agarose gels containing ethidium bromide. Bands at the expected size were excised, DNA was extracted, and the fragment was cloned into pCR4Blunt-TOPO vector (Invitrogen) and confirmed by sequencing. Other probes used for ISH have been described elsewhere (*Bouilloux et al., 2016*).

## In situ hybridization (ISH)

Before hybridization, slides were air-dried for 2–3 hr at room temperature. Plasmids containing probes were used to synthesize digoxigenin-labeled or fluorescein-labeled antisense riboprobes according to the supplier's instructions (Roche) and purified by LiCl precipitation. Sections were hybridized overnight at 70°C with a solution containing 0.19 M NaCl, 10 mM Tris (pH 7.2), 5 mM NaH$_2$PO$_4$*2H$_2$O/ Na$_2$HPO$_4$ (pH 6.8), 50 mM EDTA, 50% formamide, 10% dextran sulfate, 1 mg/ml yeast tRNA, 1× Denhardt solution, and 100–200 ng/ml of probe. Sections were then washed four times for 20 min at 65°C in 0.4× SSC pH 7.5, 50% formamide, 0.1% Tween 20, and three times for 20 min at room temperature in 0.1 M maleic acid, 0.15 M NaCl, and 0.1% Tween 20 (pH 7.5). Sections were blocked for 1 hr at room temperature in the presence of 20% goat serum and 2% blocking agent (Roche) prior to incubation overnight with AP-conjugated anti-DIG-Fab-fragments (Roche, 1:2000). After extensive washing, hybridized riboprobes were revealed by performing an NBT/BCIP reaction in 0.1 M Tris–HCl pH 9.5, 100 mM NaCl, 50 mM MgCl$_2$, and 0.1% Tween 20.

For double ISH, the procedure was the same except that hybridization was conducted by incubation with 100–200 ng/ml of one digoxigenin-labeled probe and 100–200 ng/ml of one fluorescein-labeled probe. Fluorescein-labeled probe was first revealed after overnight incubation with AP-conjugated anti-fluorescein-Fab-fragment (Roche, 1:2000) and further incubation with Fast Red tablets in 0.1 M Tris–HCl pH 8.5, 100 mM NaCl, 50 mM MgCl$_2$, and 0.1% Tween 20. Pictures of fluorescein alone

were taken after mounting in glycerol/PBS (1:9). To reveal the digoxigenin-labeled probe, sections were unmounted, washed extensively in PBS, and alkaline phosphatase was inhibited by incubation in a solution of 0.1 M glycin pH 2.2, 0.2% Tween 20 for 30 min at room temperature. After extensive washing in PBS, digoxigenin-labeled probe was revealed as described using the AP-conjugated anti-DIG-Fab-fragments (Roche, 1:2000) and the NBT/BCIP reaction. Sections were mounted again in glycerol/PBS (1:9), and pictures of both fluorescein and digoxigenin were taken. For removing the Fast Red staining, sections were unmounted again, washed extensively in PBS, and incubated in increasing solutions of ethanol/PBS solutions (20–100% ethanol). After extensive washing in PBS, sections were mounted in glycerol/PBS (1:9), and pictures of the digoxigenin staining alone were taken. Wide-field microscopy (Leica DMRB, Germany) was only used for ISH and ISH combined with immunochemistry.

## Immunochemistry

Immunochemistry was performed as previously described (*Bouilloux et al., 2016*). In situ hybridized sections or new sections were washed 3 × 10 min with PBS, blocked with 4% normal goat serum, 1% bovine serum albumin and 0.1% Triton X100 in PBS, and incubated overnight at 4°C with primary antibodies. After washing 3 × 10 min with PBS, incubation occurred for 2–4 hr with secondary species and isotype-specific fluorescent antibodies (Alexa Fluor Secondary Antibodies, Molecular Probes). After repeated washing with PBS, slides were mounted in glycerol/PBS (9/1) or Mowiol. Pictures were taken using a confocal microscope (Leica SP5-SMD, Germany). Confocal images are presented as maximal projections.

The following antibodies were used for immunochemistry: mouse anti-islet1 39.4D used for mouse and chick (diluted 1:100, Developmental Studies Hybridoma Bank); goat anti-TrkB (1/2000; R&D Systems, Cat# AF1494); goat anti-TrkC (1/1000; R&D Systems, Cat# AF1404); rabbit anti-TrkA (1/500; Millipore, Cat# 06-574); rabbit anti-parvalbumin antibody (1:500, Swant, Cat# PV 25); guinea pig anti-calcitonin gene-related peptide (CGRP) antibody (1:500, BMA Biomedicals, Cat# T-5053); rabbit anti-Nefh (Sigma, rabbit 1:1000, Cat# N4142), guinea pig anti-c-maf (generous gift of C. Birchmeier, MDC, 1/10,000), rabbit anti-TrpV1 (Sigma, Cat# V2764, 1:1000), mouse-anti-S100β (Sigma, Cat# S2532, 1:1000), rabbit anti-phospho-CREB (Cell Signaling, Cat# 87G3, 1/200, Germany); goat anti-Ret (R&D Systems, Cat# AF482, 1/100) and mouse or rabbit anti-Meis2 (Sigma-Aldrich, WH0004212M1 or Abcam ab244267, 1/500), and rat anti-CK8 (DSHB, TROMA-I, 1/100). The chick rabbit anti-TrkB and C antibodies were a generous gift from LF Reichardt, UCSF and have been previously reported.

## Whole-mount immunohistochemistry

Whole-mount immunohistochemistry of adult mice back hairy skin was performed as described elsewhere (*Chang et al., 2014*). Briefly, mice were euthanized by $CO_2$ asphyxiation. The back skin was shaved and cleaned with commercial hair remover. The back skin was removed, carefully handled with curved forceps, and fixated in 4% PFA at 4°C for 20 min. The tissue was then washed with PBS containing 0.3% Triton X-100 (PBST) every 30 min for 3–5 hr and kept overnight in the washing solution. The next day, the skin was incubated for 5 d with primary antibodies diluted in PBST containing 5% donkey serum and 20% DMSO. The skin was washed the following day 8–10 times over a day before being incubated with secondary antibodies diluted in PBST containing 5% donkey serum and 20% DMSO. The skin was then washed every 30 min for 6–8 hr before being dehydrated in successive baths of 25, 50, 75, and 100% methanol. They were then incubated overnight in a 1:2 mixture of benzyl alcohol and benzyl benzoate before being mounted and sealed into chambers filled with the same medium.

## Hematoxylin–eosin staining

As previously described (*Bouilloux et al., 2016*), air-dried frozen sections were washed in water then stained with hematoxylin for 1 min at room temperature and washed extensively with water. After dehydration in PBS/alcohol (70%), slides were stained with eosin for 30 s at room temperature. After serial washing in water, sections were dehydrated in PBS solutions with increasing alcohol concentration (50, 75, 95, and 100%), mounted and observed with a microscope (Leica DMRB, Germany).

## RNA-sequencing and analysis

DRGs were dissected from E18.5 mouse embryos, collected in lysis buffer, and stored at –80°C until RNA extraction with RNeasy extraction kit (QIAGEN). After mRNA purification using the NEBNext Poly(A) mRNA Magnetic Isolation (NEB), libraries were prepared with the CORALL mRNA-Seq Library Prep Kits with UDIs (Lexogen) following the manufacturer's recommendations. After a qPCR assay to determine the optimal PCR cycle number for endpoint PCR, 14 PCR cycles were completed to finalize the library preparation. Quantitation and quality assessment of each library were performed using Qubit 4.0 (HS DNA kit, Thermo Fisher) and 4150 Tapestation analyzer (D5000 ScreenTape kit, Agilent). Indexed libraries were sequenced in an equimolar manner on NextSeq 500 Illumina sequencer. Sequencing conditions were as follows: denatured libraries were loaded on a HighOutput flowcell kit v2 and sequenced in single-end 84 pb reads. Data were extracted and processed following Illumina recommendations. After a quality check of the fastq files with FastQC, UMI sequences were extracted with UMI tools (version 1.1.2) (*Smith et al., 2017*) default parameters followed by STAR alignment (version 2.7.10) (*Dobin et al., 2013*) on mm10 genome and removal of PCR duplicate with UMI tools, default parameters. Uniquely mapped sequences from the STAR output files (bam format) were then used for further analysis. HT-seq count (version 0.6) (*Anders et al., 2015*) was used to aggregate read count per gene followed by differential gene expression analysis with Limma voom on Galaxy (version 3.50.1) (*Law et al., 2014*). Data are available under the following accession codes: GSE223788. Only genes that exhibited more than 100 reads in any of the samples were kept in the analysis. Genes with more than 1.2-fold differential expression and p-value<0.05 were used for GO analysis (https://david.ncifcrf.gov) using the list of expressed genes in our experiment as background.

## Mouse skin-nerve preparation and sensory afferent recordings

Cutaneous sensory fiber recordings were performed using the ex vivo skin-nerve preparation as previously described (*Schwaller et al., 2021*). Mice were euthanized by $CO_2$ inhalation for 2–4 min followed by cervical dislocation. Three different preparations were performed in separate experiments using different paw regions: the saphenous nerve innervating the hairy hind paw skin; the tibial nerve innervating the glabrous hind paw skin; and the medial and ulnar nerves innervating the forepaw glabrous skin. In all preparations, the hairy skin of the limb was shaved and the skin and nerve were dissected free and transferred to the recording chamber, where muscle, bone, and tendon tissues were removed from the skin to improve recording quality. The recording chamber was perfused with a 32°C synthetic interstitial fluid: 123 mM NaCl, 3.5 mM KCl, 0.7 mM $MgSO_4$, 1.7 mM $NaH_2PO_4$, 2.0 mM $CaCl_2$, 9.5 mM sodium gluconate, 5.5 mM glucose, 7.5 mM sucrose, and 10 mM 4-(2-hydroxyethyl)-1-piperazine-etha nesulfonic acid (HEPES), pH 7.4. The skin was pinned out and stretched, such that the outside of the skin could be stimulated using stimulator probes. The peripheral nerve was fed through an adjacent chamber in mineral oil, where fine filaments were teased from the nerve and placed on a silver-wire recording electrode.

The receptive fields of individual mechanoreceptors were identified by mechanically probing the surface of the skin with a blunt glass rod or blunt forceps. Analog output from a Neurolog amplifier was filtered and digitized using the Powerlab 4/30 system and Labchart 7.1 software (AD Instruments). Spike-histogram extension for Labchart 7.1 was used to sort spikes of individual units. Electrical stimuli (1 Hz, square pulses of 50–500 ms) were delivered to single-unit receptive fields to measure conduction velocity and enable classification as C-fibers (velocity < 1.2 m s$^{-1}$), Aδ-fibers (1.2–10 m s$^{-1}$), or Aβ-fibers (>10 m s$^{-1}$). Mechanical stimulation of the receptive fields of neurons was performed using a piezo actuator (Physik Instrumente, Cat# P-841.60) and a double-ended Nanomotor (Kleindiek Nanotechnik, Cat# MM-NM3108) connected to a force measurement device (Kleindiek Nanotechnik, Cat# PL-FMS-LS). Calibrated force measurements were acquired simultaneously using the Powerlab system and Labchart software during the experiment.

As different fiber types have different stimulus-tuning properties, different mechanical stimuli protocols were used based on the unit type. Low-threshold Aβ-fibers (RAMs and SAMs) and Aδ-fiber D-hairs were stimulated with the piezo actuator with three vibration stimuli (5 Hz, 25 Hz, and 50 Hz, distortions introduced by the in-series force sensor precluded using frequencies >50 Hz) with increasing amplitude over six steps (peak-to-peak amplitudes of ~6–65 mN; 20 cycles per step), and a dynamic stimulus sequence with four ramp-and-hold waveforms with varying probe deflection velocities (3 s duration; 0.075, 0.15, 0.45, and 1.5 mm s$^{-1}$; average amplitude 100 mN). Aβ-fiber SAMs and

RAMs were classified by the presence or absence of firing during the static phase of a ramp-and-hold stimulus, respectively, as previously described. Single units were additionally stimulated with a series of five static mechanical stimuli with ramp-and-hold waveforms of increasing amplitude (3 s duration; ranging from ~10 mN to 260 mN). Low-threshold SAMs, high-threshold Aδ-fibers, and C-fibers were also stimulated using the nanomotor with five ramp-and-hold stimuli with increasing amplitudes.

## Behavioral assays

### Von Frey paw withdrawal test
Mice were placed on an elevated wire mesh grid into PVC chambers. Before the test, mice were habituated to the device for 1 hr for two consecutive days. On the testing day, mice were placed in the chamber 1 hr before Von Frey filaments application. The test was performed as previously described (*Neubarth et al., 2020*). During the test, withdrawal response following Von Frey filament application on the palm of the left hind paw was measured. Starting with the lowest force, each filament ranging from 0.008 g to 1.4 g was applied 10 times in a row with a break of 30 s following the fifth application. During each application, bend filament was maintained for 4–5 s. The number of paw withdrawals for each filament was counted.

### Hot plate test
Before starting the test, mice were habituated to the experimentation room for at least 5 min. Mice were individually placed on the hot plate set up at 53°C and removed at the first signs of aversive behavior (paw licking or shaking). The time to this first stimulus was recorded. A 30 s cutoff was applied to avoid skin damage. After 5 min recovery in their home cage, the test was repeated three times for each mouse and averaged. Data are shown as the average of these three measurements.

### Sticky tape test
A 2 cm$^2$ of laboratory tape was placed on the upper-back skin of mice just before they were placed on an elevated wire mesh grid into PVC chambers. The number of tape-directed reactions was then counted during 5 min. Considered responses were body shaking like a 'wet dog', hindlimb scratching directed to the tape, trying to reach the tape with the snout and grooming of the neck with forepaws.

### Dynamic touch test
Mice were placed in the same conditions as described above for the Von Frey paw withdrawal test. Sensitivity to dynamic touch was performed by stroking hind paws with a tapered cotton swab in a heel-to-toe direction. The stimulation was repeated 10 times by alternating left and right hind paws and the number of paw withdrawals was counted.

### Gait analysis
Gait was analyzed using the Catwalk system (Noldus Information Technology, the Netherlands) in a dark room with minimized light emission from the computer screen. Mice were allowed to voluntarily cross a 100-cm-long, 5-cm-wide walkway with a glass platform illuminated by green fluorescent light. An illuminated image is produced when a mouse paw touches the glass floor through dispersion of the green light, and footprints were captured by a high-speed camera placed under the glass floor. Data were analyzed using the CatWalk XT 10.1 software. For each mouse, several recordings were performed until at least three runs met the criteria defined by a minimum of three consecutive complete step cycles of all four paws without stopping or hesitation and within the range of 25–50 cm s$^{-1}$. Data are reported as the average of at least three runs per mouse.

### Materials availability
Probes generated for ISH are available upon request to the corresponding author. Meis2$^{LoxP/LoxP}$ mice strain is available upon request to the corresponding author by signing a material transfer agreement (MTA). Sequencing data are available under the following accession code: GSE223788.

## Acknowledgements

We thank Stéphanie Ventéo for help with ChTx experiments, Anne-Laure Bonnefont for catwalk experiments, and all staff at animal house and in particular Flora for great help at the animal facility. We also thank Yves Dusabyinema, Benjamin Gillet, and Sandrine Hughes at the IGFL sequencing platform (PSI) for Illumina sequencing.

## Additional information

### Funding
No external funding was received for this work.

### Author contributions
Simon Desiderio, Frederick Schwaller, Formal analysis, Investigation, Methodology; Kevin Tartour, Formal analysis; Kiran Padmanabhan, Funding acquisition, Project administration, Writing – review and editing; Gary R Lewin, Data curation, Formal analysis, Methodology, Writing – review and editing; Patrick Carroll, Funding acquisition, Writing – review and editing; Frederic Marmigere, Conceptualization, Data curation, Formal analysis, Supervision, Funding acquisition, Investigation, Methodology, Writing – original draft, Project administration, Writing – review and editing

### Author ORCIDs
Frederick Schwaller ⓘ https://orcid.org/0000-0002-7685-5537
Kiran Padmanabhan ⓘ https://orcid.org/0000-0001-7020-1682
Gary R Lewin ⓘ http://orcid.org/0000-0002-2890-6352
Patrick Carroll ⓘ http://orcid.org/0000-0003-3287-8195
Frederic Marmigere ⓘ https://orcid.org/0000-0002-0515-7483

### Ethics
All procedures involving animals and their care were conducted according to European Parliament Directive 2010/63/EU and the 22 September 2010 Council on the protection of animals, and were approved by the French Ministry of research (APAFIS#17869-2018112914501928 v2, June the 4th of 2020).

Joint Public Review: https://doi.org/10.7554/eLife.89287.3.sa1
Author Response https://doi.org/10.7554/eLife.89287.3.sa2

## Additional files

### Supplementary files
• MDAR checklist

### Data availability
Sequencing data have been deposited in GEO under accession codes GSE223788.

The following dataset was generated:

| Author(s) | Year | Dataset title | Dataset URL | Database and Identifier |
|---|---|---|---|---|
| Tartour K, Carroll P, Padmanabhan K, Marmigère F | 2024 | Touch receptor end-organ innervation and function requires sensory expression of the transcription factor Meis2 | http://www.ncbi.nlm.nih.gov/geo/query/acc.cgi?acc=GSE223788 | NCBI Gene Expression Omnibus, GSE223788 |

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
