## [Editor Report · eLife assessment]

This **fundamental** study identifies the homeodomain transcription factor Meis2 as a transcriptional regulator of maturation and end-organ innervation of low-threshold mechanoreceptors (LTMRs) in the dorsal root ganglia (DRG) of mice. The authors use histology, behavioral tests, RNA-sequencing, and electrophysiological recordings to provide evidence that conditional deletion of Meis2 in postmitotic DRG neurons causes gene expression changes together with targeting errors and altered sensory neuron responses, ultimately resulting in reduced sensitivity to light touch in mutant animals. The data presented are **convincing**, the discussion comprehensive, and the conclusions drawn justified.

---

## [Referee Report · Joint Public Review]

Summary:

Desiderio and colleagues investigated the role of the TALE (three amino acid loop extension) homeodomain transcription factor Meis2 during maturation and target innervation of mechanoreceptors and their sensation to touch. They start with a series of careful in situ hybridizations and immunohistochemical analyses to examine Meis2 transcript expression and protein distribution in mouse and chick DRGs of different embryonic stages. By this approach, they identify Meis2+ neurons as slowly- and rapidly adapting A-beta LTMRs, respectively. Retrograde tracing experiments in newborn mice confirmed that Meis2-expressing sensory neurons project to the skin, while unilateral limb bud ablations in chick embryos in ovo showed that these neurons require target-derived signals for survival. The authors further generated a conditional knock-out (cKO) mouse model in which Meis2 is selectively lost in Islet1-expressing, postmitotic neurons in the DRG (IsletCre/+::Meis2flox/flox, abbreviated below as cKO). WT and Islet1Cre/+ littermates served as controls. cKO mice did not exhibit any obvious alteration in volume or cellular composition of the DRGs but showed significantly reduced sensitivity to touch stimuli and various innervation defects to different end-organ targets. RNA-sequencing experiments of E18.5 DRGs taken from WT, Islet1Cre/+ and cKO mice reveals extensive gene expression differences between cKO cells and the two controls, including synaptic proteins and components of GABAergic- and glutamatergic transmission. Histological analysis and electrophysiological recordings shed light on the physiological defects resulting from the loss of Meis2. By immunohistochemical approaches, the authors describe distinct innervation defects in glabrous and hairy skin (reduced innervation of Merkel cells by SA1-LTMRs in glabrous but not hairy skin, reduced complexity of A-beta RA1-LTMs innervating Meissner's corpuscles in glabrous skin, reduced branching and innervation of A-betA RA1-LTMRs in hairy skin). Electrophysiological recordings from ex vivo skin nerve preparations found that several, but not all of these histological defects are matched by altered responses to external stimuli, indicating that compensation may play a considerable role in this system. This study will be of interest to developmental biologists and neuroscientists, in particular those interested in the sensation of touch.

Strengths:

This is a well-conducted study that combines different experimental approaches to convincingly show that the transcription factor Meis2 plays an important role in the perception of light touch. The authors describe a new mouse model for compromised touch sensation, characterize it by histology and electrophysiological recordings, and identify several genes whose expression depends on Meis2 in mouse DRGs.

Weaknesses:

The authors use different experimental approaches to investigate the role of Meis2 in touch sensation, but the results obtained by these techniques could be better connected. For instance, the authors identify several genes involved in synapse formation, synaptic transmission, neuronal projections, or axon and dendrite maturation that are up- or downregulated upon targeted Meis2 deletion, but it remains to be resolved whether these chances explain the histological, electrophysiological, or behavioral deficits observed in cKO animals.

---

## [Author Response]

The following is the authors’ response to the original reviews.

**eLife assessment**
This fundamental study identifies the homeodomain transcription factor and suspected autism-candidate gene Meis2 as transcriptional regulators of maturation and end-organ innervation of low-threshold mechanoreceptors (LTMRs) in the dorsal root ganglia (DRG) of mice. For a few years, the view on autism spectrum disorders (ASD) has shifted from a disorder that exclusively affects the brain to a condition that also includes the peripheral somatosensory system, even though our knowledge about the genes involved is incomplete. The study by Desiderio and colleagues is therefore not only scientifically interesting but may also have clinical relevance. The work is convincing, with appropriate and validated methodology in line with current state-of-the-art and the findings contribute both to understanding and potential application.
**Public Reviews:**

**Reviewer #1 (Public Review):**
Summary:This work examined transcription factor Meis2 in the development of mouse and chick DRG neurons, using a combination of techniques, such as the generation of a new conditional mutant strain of Meis2, behavioral assays, in situ hybridization, transcriptomic study, immunohistochemistry, and electrophysiological (ex vivo skin-nerve preparation) recordings. The authors found that Meis2 was selectively expressed in A fiber LTMRs and that its disruption affects the A-LTMRs' end-organ innervation, transcriptome, electrophysiological properties, and light touch-sensation.Strengths:1. The authors utilized a well-designed mouse genetics strategy to generate a mouse model where the Meis2 is selectively ablated from pre- and post-mitotic mouse DRG neurons. They used a combination of readouts, such as in situ hybridization, immunhistochemistry, transcriptomic analysis, skin-nerve preparation, electrophysiological recordings, and behavioral assays to determine the role of Meis2 in mouse DRG afferents.2. They observed a similar preferential expression of Meis2 in large-diameter DRG neurons during development in chicken, suggesting evolutionarily conserved functions of this transcription factor.3. Conducted severe behavioral assays to probe the reduction of light-touch sensitivity in mouse glabrous and hairy skin. Their behavioral findings support the idea that the function of Meis2 is essential for the development and/or maturation of LTMRs.4. RNAseq data provide potential molecular pathways through which Meis2 regulates embryonic target-field innervation.5. Well-performed electrophysiological study using skin-nerve preparation and recordings from saphenous and tibial nerves to investigate physiological deficits of Meis2 mutant sensory afferents.6. Nice whole-mount IHC of the hair skin, convincingly showing morphological deficits of Meis2 mutant SA- and RA- LTMRs.Overall, this manuscript is well-written. The experimental design and data quality are good, and the conclusion from the experimental results is logical.Weaknesses:1. Although the authors justify this study for the involvement of Meis2 in Autism and Autism associated disorders, no experiments really investigated Autism-like specific behavior in the Meis2 ablated mice.

Indeed, in the first version of the manuscript, we use current understanding of ASD in mouse models and associated sensory defects to articulate our introduction and discussion. As noticed by reviewer 1, none of our experiments really investigated ASD. To avoid over-interpretation of the data, we have now removed sentences mentioning ASD and related references throughout the manuscript.

2. For mechanical force sensing-related behavioral assays, the authors performed VFH and dynamic cotton swabs for the glabrous skin, and sticky tape on the back (hairy skin) for the hairy skin. A few additional experiments involving glabrous skin plantar surfaces, such as stick tape or flow texture discrimination, would make the conclusion stronger.

We fully agree on that performing more behavioral analysis investigating with more details the primary sensory defects as well as some ASD-related behavior would re-inforce our conclusions. Our behavioral analysis clearly showed a loss of sensitivity in response to mechanical stimuli within the light touch range but not for higher range mechanical or noxious thermal stimuli. While the experiments suggested by the reviewer are interesting and would strengthen our conclusions, they are far from trivial and require large cohorts. Given the current laboratory conditions as stated at the outset, these unfortunately are not within reach.

3. The authors considered von Frey filaments (1 and 1.4 g) as noxious mechanical stimuli (Figure 1E and statement on lines 181-183), which is questionable. Alligator clips or pinpricks are more certain to activate mechanical nociceptors.

To avoid misinterpretation of the higher Von Frey filament tests, we deleted the two following statement in page 7: “In the von Frey test, the thresholds for paw withdrawal were similar between all genotypes when using filaments exerting forces ranging from 1 to 1.4g, which likely reflects the activation of mechanical nociception suggesting that Meis2 gene inactivation did not affect nociceptor function.”. The sentence “… while sparing other somatosensory behaviors” was also deleted.

4. There are disconnections and inconsistencies among findings from morphological characterization, physiological recordings, and behavior assays. For example, Meis2 mutant SA-LTMRs show a deficiency in Merkel cell innervation in the glabrous skin but not in hairy skin. With no clear justification, the authors pooled recordings of SA-LTMRs from both glabrous and hairy skin and found a significant increase in mean vibration threshold. Will the results be significantly different if the data are analyzed separately? In addition, whole-mount IHC of Meissner's corpuscles showed morphological changes, but electrophysiological recordings didn't find significant alternation of RAI LTMRs. What does the morphological change mean then? Since the authors found that Meis2 mice are less sensitive to a dynamic cotton swab, which is usually considered as an RA-LTMR mediated behavior, is the SAI-LTMR deficit here responsible for this behavior? Connections among results from different methods are not clear, and the inconsistency should be discussed.

We thank Reviewer 1 for the careful review of our data and fully agree with the weaknesses identified, weaknesses we were ourselves aware of at the time of submission. In particular on the lack of stronger connections between histological and electrophysiological data. Electrophysiological studies were conducted on a first cohort of mice where we mostly emphasize on WT and Meis2 mutant mice. The goal was to describe differences in electrophysiological properties of identified mechanoreceptors from these two genotypes. While substantial differences between WT and Islet1-Cre mice were not expected, only very few mice with this genotype were examined at that time to confirm this assumption. We fully agree with reviewer 1 that confirming differences in SA-LTMRs responses in the hairy and glabrous at electrophysiological levels would be interesting and worthwhile. It is assumed that the physiological properties of SA-LTMRs from glabrous and hairy skins are equivalent in both skin types. Indeed direct comparisons have been made between glabrous and hairy skin SA-LTMRs revealing that they have equivalent receptor properties (see Walcher et al J Physiol quoted in the manuscript). We had not recorded from a sufficient number of hairy and glabrous skin SA-LTMRs to make any meaningful comparison statistically. When we noticed the dramatic differences in the innervation patterns of Merkel cell complexes between glabrous and hairy skin, we immediately planned a second mice cohort, but as explained in the onset to the Public Review, this cohort was sacrificed due to the pandemic lockdown. However, the obtained dataset clearly shows that in Meis2 mutant mice many SA-LTMRs had similar vibration thresholds to those of wild types.

For Meissner corpuscle, histological analysis evidenced clear morphological differences that could of course be investigated at the level of the dual innervation previously reported by Neubarth et al. It is uncertain whether differences in their electrophysiological responses would be revealed by increasing the number of recorded fibers. For this reason, we clearly stated this limitation in the results section page 7 “There was a tendency for RA-LTMRs in Isl1Cre/+::Meis2LoxP/LoxP mutant mice to fire fewer action potentials to sinusoids and to the ramp phase of a series 2 second duration ramp and hold stimuli, but these differences were not statistically significant (Figure 5B). Nevertheless it is important to point out that an electrical search strategy revealed that many Aβ-fibers did not have mechanosensitive receptive fields. Thus by focusing on LTMRs with a mechanosensitive receptive field, we ignore the fact that fewer fibers are mechanosensitive. This is now more extensively discussed in the discussion section of the manuscript page 13:

“Indeed, the electrophysiology methods used here can only identify sensory afferents that have a mechanosensitive receptive field. Primary afferents that have an axon in the skin but no mechanosensitvity can only be identified with a so-called electrical search protocol (45, 46) which was not used here. It is therefore quite likely that many primary afferents that failed to form endings would not be recorded in these experiments e.g. SA-LTMRs and RA-LTMRs that fail to innervate end-organs (Fig.4-6).”

“From our data, we could not conclude whether SA-LTMR electrophysiological responses are differentially affected in the glabrous versus hairy skin of Meis2 mutant as suggested by histological analysis. Further electrophysiological analysis focused on SA-LTMR selectively innervating the glabrous or hairy skin would be necessary to answer this question. Similarly, the decreased sensitivity of Meis2 mutant mice in the cotton swab assay and the morphological defects of Meissner corpuscles evidenced in histological analysis do not correlate with RA-LTMR electrophysiological responses for which a tendency to decreased responses were however measured. The later might result from an insufficient number of fibers recording, whereas the first may be due of pooling SA-LTMR from both the hairy and glabrous skin.”.

**Reviewer #2 (Public Review):**
Summary:Desiderio and colleagues investigated the role of the TALE (three amino acid loop extension) homeodomain transcription factor Meis2 during maturation and target innervation of mechanoreceptors and their sensation to touch. They start with a series of careful in situ hybridizations to examine Meis2 transcript expression in mouse and chick DRGs of different embryonic stages. By this approach, they identify Meis2+ neurons as slowly- and rapidly adapting A-beta LTMRs, respectively. Retrograde tracing experiments in newborn mice confirmed that Meis2-expressing sensory neurons project to the skin, while unilateral limb bud ablations in chick embryos in Ovo showed that these neurons require target-derived signals for survival. The authors further generated a conditional knock-out (cKO) mouse model in which Meis2 is selectively lost in Islet1-expressing, postmitotic neurons in the DRG (IsletCre/+::Meis2flox/flox, abbreviated below as cKO). WT and Islet1Cre/+ littermates served as controls. cKO mice did not exhibit any obvious alteration in volume or cellular composition of the DRGs but showed significantly reduced sensitivity to touch stimuli and various innervation defects to different end-organ targets. RNA-sequencing experiments of E18.5 DRGs taken from WT, Islet1Cre/+, and cKO mice reveal extensive gene expression differences between cKO cells and the two controls, including synaptic proteins and components of the GABAergic signaling system. Gene expression also differed considerably between WT and heterozygous Islet1Cre/+ mice while several of the other parameters tested did not. These findings suggest that Islet1 heterozygosity affects gene expression in sensory neurons but not sensory neuron functionality. However, only some of the parameters tested were assessed for all three genotypes. Histological analysis and electrophysiological recordings shed light on the physiological defects resulting from the loss of Meis2. By immunohistochemical approaches, the authors describe distinct innervation defects in glabrous and hairy skin (reduced innervation of Merkel cells by SA1-LTMRs in glabrous but not hairy skin, reduced complexity of A-beta RA1-LTMs innervating Meissner's corpuscles in glabrous skin, reduced branching and innervation of A-betA RA1-LTMRs in hairy skin). Electrophysiological recordings from ex vivo skin nerve preparations found that several, but not all of these histological defects are matched by altered responses to external stimuli, indicating that compensation may play a considerable role in this system.Strengths:This is a well-conducted study that combines different experimental approaches to convincingly show that the transcription factor Meis2 plays an important role in the perception of light touch. The authors describe a new mouse model for compromised touch sensation and identify a number of genes whose expression depends on Meis2 in mouse DRGs. Given that dysbalanced MEIS2 expression in humans has been linked to autism and that autism seems to involve an inappropriate response to light touch, the present study makes a novel and important link between this gene and ASD.Weaknesses:The authors make use of different experimental approaches to investigate the role of Meis2 in touch sensation, but the results obtained by these techniques could be connected better. For instance, the authors identify several genes involved in synapse formation, synaptic transmission, neuronal projections, or axon and dendrite maturation that are up- or downregulated upon targeted Meis2 deletion, but it is unresolved whether these chances can in any way explain the histological, electrophysiological, or behavioral deficits observed in cKO animals. The use of two different controls (WT and Islet1Cre/+) is unsatisfactory and it is not clear why some parameters were studied in all three genotypes (WT, Islet1Cre/+ and cKO) and others only in WT and cKO. In addition, Meis2 mutant mice apparently are less responsive to touch, whereas in humans, mutation or genomic deletion involving the MEIS2 gene locus is associated with ASD, a condition that, if anything, is associated with an elevated sensitivity to touch. It would be interesting to know how the authors reconcile these two findings. A minor weakness, the first manuscript suffers from some ambiguities and errors, but these can be easily corrected.

We thank the reviewer for the insightful comments and suggestions.

The use of two different controls (WT and Islet1Cre/+) is unsatisfactory and it is not clear why some parameters were studied in all three genotypes (WT, Islet1Cre/+ and cKO) and others only in WT and cKO.

First, we identified a labelling mistake in figures 4D, 5A and 6A where the control shown are from Islet1+/Cre mice and not from WT as reported in the first version. We apologize for this mistake which has now been corrected. This typographical error does not in any way affect our conclusion, on the contrary, it shows that innervation defects are not the consequence of Islet1 heterozygosity.

The reviewer wonders why for some data both control genotypes are presented, and for some others only one is presented. It is quite possible that genes expression changes happen due to a synergistic effect of both heterozygous Meis2 deletion and heterozygous Islet1 deletion. However, we found no evidence that this led to defects in target-field innervation or to changes in the physiological properties of sensory neurons.

Whereas it could be fairly envisaged that some gene expression is modified due to a synergistic effect of both heterozygous Meis2 deletion and heterozygous deletion of Islet1, several lines of evidence support that the defects in target-field innervation and electrophysiological responses are exclusively due to Meis2 deletion.Previous work on Islet1 specific deletion in DRG sensory neurons opens the possibility that some of the phenotypes we report here are in part due to an effect of Islet1 heterozygous deletion or a synergistic effect to Meis2 homozygous deletion.

1. When Islet1 is conditionally deleted in mice using the Wnt1-Cre strain or at later stages using a tamoxifen inducible-Cre, homozygous pups die a few hours after birth. Early Islet1 deletion results in an increased apoptosis in the DRG, a massive loss of DRG sensory neurons and sensory defects associated to nociceptors mostly and some touch neurons while proprioceptive neurons are spared (Sun et al., 2008 now included in the revised version of the manuscript). There was a decrease in the number of Ntrk1+ and Ntrk2+ neurons whereas Ntrk3+ neurons number appeared normal. When Islet1 is inactivated later in development, the number of Ntrk1+ and Ntrk2+ neurons were normal and only the expression of nociceptor specific markers was decreased. Since neither the DRG volume, nor the number of Ntrk1+, Ntrk2+ and Ntrk3+ neurons are changed in Meis2 cKO using the Islet1-Cre strain, an early significant effect of Islet1 heterozygous deletion is very unlikely.

2. For distal innervation defects, it is clear from the Wnt1-Cre::Meis2 data (Figure 3E) that the distal innervation phenotype occurred while Meis2 is inactivated independently of Islet1 expression.

3. Finally, the lack of differences between WT and Islet+/Cre mice in behavioral assays and in electrophysiological characterization of RA-LTMR of the hairy skin (Figure 6C) and SA-LTMR (Figure 4B and C) argues for a lack of significant consequences of Islet1 heterozygous deletion on these parameters.

4. For bulk RNAseq studies, all datasets has been now re-analyzed following Reviewer 2 specific comments (see below). To avoid misinterpretation of the data, the results are now presented differently (see pages 8 and 9) and more critically discussed (see pages 14 and 15). In particular, we included and discuss references on Islet1 cKO mice.

We also agree with reviewer 2 that our RNAseq study only provides cues on potential genes expression that could impact distal innervation and electrophysiological responses. However, proving which of those genes are fully responsible for the morphological and electrophysiological defects would require extensive mouse genetic investigations such as restoring their normal expression level in a Meis2 mutant context, which is beyond the scope of the present study.

Finally, the reviewer questioned how we could reconcile the lower touch sensitivity in Meis2 mutant mice with the exacerbated touch sensitivity found in ASD patient and mouse models of ASD. As suggested by reviewer 1, our study did not really investigate ASD specifically. Therefore, to avoid over interpretation of the data and to follow Reviewer 1 recommendation, we have removed all references to ASD in the revised version of the manuscript. Indeed, to our knowledge, none of the case reports on Meis2 mutant patients investigated sensory function in general and light touch in particular, maybe because of the severe intellectual disability characterizing these patients.

**Reviewer #1 (Recommendations For The Authors):**
In addition to the aforesaid suggestions in the section 2, there are some minor issues:

We thank the reviewer for the careful reading and for identifying all these typos. All of them have been corrected in the revised version of the manuscript.

1. There should not be a full stop mark in the title of the article.This has been corrected in the new version of the manuscript.2. Figure 1C, 1D, please correct the typo "controlateral' to "contralateral".

This has been corrected in the new version of the manuscript.

3. Figure 1D, lower graph, Y-axis, please correct the typo 'umber' to "number".

This has been corrected in the new version of the manuscript.

4. To make it easy for readers, add the names of the behavioral tests on top of the graphs in Fig 1E-H.

The name of behavioral tests is now added to the figure.

5. It would be easier to read the markers' names in IHC and ISH images if they were written outside of image panels. The blue staining color in image 1B could be easily mixed with the background. Suggest change colors.

Markers for IHC and IH images are now written outside the image panel or colors have been change in figure 1 and 2 for better clarity.

6. The font size of Genes' name in Figure 3B is too small and not readable.

Figure 3 has now been changed following Reviewer 2 recommendation. The small font size in Figure 3B is no longer present in the figure.

7. Quantification of Fig 3E (number of fibers innervating each dermal papilla or footpad, for example).

Unfortunately, we did not kept the Wnt1Cre::Meis2LoxP/LoxP strain which prevents further analysis (see onset of the answer to public review).

8. In Figure 4, please arrange IHC images and their quantification results adjacent to each other.

The figure has been reorganized and changes in the result section and figures legend were made accordingly.

9. For consistency, please use either LTMR or LTM (See Figure 4F, 5A, 6C), but not both.

This has been homogenized throughout the manuscript.

10. Add arrows/heads to mark the overlaps in Figure 4D.

Arrows are now added in Figure 4D to point at the overlap between Nefh and CK8 staining.

11. Figure 5A, 6A, Lines 236, 240, 247, 258, 305, 308, 313, 347, and many more in Figure legends: please check in entire manuscript and make the mouse genotype nomenclature (+/Cre?) consistent. In some places, Cre is written in all upper case (Line 657).

This has been homogenized throughout the manuscript.

12. Figure 4G: Histogram color could be darker for better contrast.

The color of the histograms has been changes in figures 6 and 5 for better clarity.

13. Please add the figure number to the Figure 6.

The figure number is now indicated on the figure.

14. Figure 6B: Y-axis typo, correct "Nfeh" to Nefh.

This typo is now corrected.

15. Either explain Figure 2B information before that of Figure 2C (In lines 204-207) in the text or change the figure panel sequence to keep the consistent flow of contents.

The figure has been modified and the panel sequence now follows that of the main text.

16. Line 213 has a typo: change "form" to "from".

This typo is now corrected.

17. Line 423 has a typo. Correct "al" to "all".

This typo is now corrected.

18. Line 625 has a typo. Correct "fo" to "of".

This typo is now corrected.

19. Line 669 has a typo. Correct "Alexa Fluo" to "Fluor".

This typo is now corrected.

20. Line 744: To be consistent in the entire manuscript, write "Nfh" as "Nefh".

This typo is now corrected.

21. 740-749: Please add host names for all primary antibodies, as some are given but some are not for the current version.

We now indicated the host species for all primary antibodies used in the study.

22. Line 751 has a typo: change "a" to "as".

This typo is now corrected.

23. Line 754: what is for 20'?

This typo is now corrected.

24. Line 832: change "day test" to "testing day".

The change has been made.

25. Please mention for how many seconds the VFH was administered on the plantar surface in the method.

A new sentence has been added to the “Von Frey withdrawal test” Methods section (page 30): “During each application, bend filament was maintained for approximately four to five seconds”.

26. For the sticky tape test, in lieu of hind paw attending bouts, wet-dog shake behavior, the authors also found some scratching behaviors. Did they separately quantify these behaviors? It would be interesting to see exactly which behavior significantly reduced after Meis2 inactivation.

Unfortunately, at the time of the design of the sticky tape test, we did not consider separating the behaviors considered as “positive” reactions. As these experiments were not video recorded, we are not able to extract this kind of information without generating new mice cohort and repeating this experiment.

27. Line 344-345: consider rephrasing the sentence.

This sentence has been removed.

**Reviewer #2 (Recommendations For The Authors):**
This is a beautiful and well-conducted study with all the strengths listed in the paragraphs above. Nevertheless, there are still some open questions, ambiguities in the presentation, and minor errors that I would recommend addressing.Major Points:1. The authors performed RNA-seq analysis from E18.5 mouse total DEGs from three different genotypes, WT, Isle1Cre/+ and cKO. Although this approach identified several interesting Meis2-dependent candidate genes, the presentation of the results is confusing, and the publication would gain impact if the RNA-seq results were better connected to the histological, behavioral, and electrophysiological data. Specific concerns:(1.1) The gene expression profiles of WT and Islet1Cre/+ samples are remarkably divergent. According to Yang Development 2006, Islet1-Cre was generated by knocking in Cre into the endogenous Islet1 locus and replacing the Isl1 ATG, hence resulting in a heterozygous null for Islet1. When purely technical derivations can be excluded, the RNAseq results presented here suggest that heterozygous loss of Islet1 causes considerable gene expression changes in the postnatal DRG. For analysis of the RNAseq results, the authors focus on genes that are differentially expressed between one experimental condition (Islet1Cre/+::Meis2flox/flox) and either one of two controls (WT or Islet1Cre/+). Hence, they pool the genes that are differently expressed between cKO and Islet1Cre/+ with the genes that are different between cKO and WT. This approach mixes gene expression differences that result from two different genetic alterations, heterozygosity of Islet1 and targeted deletion of Meis2, respectively. It seems much more logical to compare the results pairwise.

We agree with reviewer 2 that heterozygous deletion of Islet1 causes a significant change in genes expression that seems to very little correlate with any of the phenotypes we investigated in the study. When Islet1 is conditionally deleted in mouse using the Wnt1-cre strain, pups die few hours after birth and display increased apoptosis in the DRG, massive loss of DRG sensory neurons and sensory defects associated to nociceptors mostly and some touch neurons while proprioceptive neurons are spared (Sun et al., 2008 now included in the revised version of the manuscript). There is a decrease numbers of Ntrk1+ and Ntrk2+ neurons whereas the numbers of Ntrk3+ neurons appear normal. Later Isl1 inactivation does not induces changes in number of neurons and does not change Ntrk1 and 2 expressions.As explained in the answer to public reviews, bulk RNAseq data have now been reanalyzed following the reviewer suggestions and presented accordingly in the related figures.

In the study bay Sun et al. they also reported DEGs following Islet1 homozygous deletion, but data on Islet1 heterozygous deletion are not included. However, out of the 60 most dysregulated genes identified in their study, only 6 were differentially expressed in our datasets. Importantly, DEGs in their studies where identified using microarray. In another study, the same group, showed that Brn3a (another transcription factor important for DRG neurons differentiation) and Islet1 exhibit negative epistasis on sensory genes expression (Dykes et al., 2011 now included in the revised version of the manuscript). Thus we cannot rule out that similar rules apply for Islet1 and Meis2. However, given the high diversity of DRG sensory neurons, interpreting our bulk RNAseq analysis in such direction might lead to misinterpretation.

(1.2) Along the same line, gene expression changes in Islet1Cre/+ DRGs seem to have little functional consequences, at least in the cases where all three genotypes were analyzed (target dependency (Fig. 1E), behavior (Fig. 1F), innervation (Fig. 4F, 6C)). Why were some parameters measured in all three genotypes and others only for WT and cKO? The authors probably reason that parameters that do not differ between WT and cKO animals will likely also not differ between WT and Islet1Cre/+. But what about parameters that do differ? Considering that the innervation of Merkel cells (Fig. 4E) and Meissner corpuscles (Fig. 5A) differ profoundly between WT and cKO, it would be interesting to know what this innervation looks like in Islet1Cre/+ DRGs. NEFH staining together with CK8 or S100beta from existing tissue sections should easily answer this question.

As explained in the answer for public reviews, there was a mistake in the annotation of the control in figure 4 D and E, and in Fig. 5 that has now been corrected. Concerning target-dependency, those are experiments conducted in chick embryo, and therefore no associated genotype.

(1.3) Was a minimum cut-off for gene expression applied? The up-and downregulated genes in Fig. 3B list a number of pseudogenes and predicted genes. A quick (and incomplete) check for their expression in Fig2 Supple Table 1 shows that only a few reads were detected for most of them. With such low expression, even small changes will show up as significant differences.

In our first analysis, a cut-off of 10 reads was applied. As reviewer 2 mentioned, this cut-off included several pseudogenes and predicted genes with low expression for which small changes were significant. We now re-analyzed the dataset using a cut-off of 100 reads. This excluded most of the previous predicted genes and pseudogenes for the analysis and resulted in a much small number of DEGs for each dataset. As recommended by reviewer 2, we also now performed the David analysis separately. These results are now presented in Figure 3 and corresponding supplementary figures.

(1.4) Given that bulk RNAseq from whole embryonic DRGs was performed, it would be interesting to know what cell type(s) express the Meis2-dependent transcripts. To address this question, the authors resort to published scRNAseq data by Usoskin Nat Neurosci 2015. They correlate the expression of all 488 DEGs (different between cKO and either WT or Islet1Cre/+) with the expression of Meis2 in the sensory neuron subtypes that were classified in the Usoskin paper. From that they conclude that many Meis2-dependent genes were expressed in the same sensory neuron classes as Meis2 itself. This is not apparent from Fig. 3 Supplementary 2. Neither do the 488 DEGs seem to be in any way enriched in the MEIS2-expressing cell clusters NF2/3/4/5, nor is cluster PEP1 particularly high in Meis2 expression. Immunostaining for MEIS2 together with a few selected DEGs would be a better way to assess co-expression.

We agree with reviewer 2 that the correlation between DEGs and the expression of Meis2 in the sensory neuron subtypes was far from striking. In our opinion, the new analysis shows now a more robust correlation. However, it has to be kept in mind that among DEGs not all are expected to be Meis2 direct target genes and therefore to be enriched in the same Meis2-expressing population. This also hold true for genes that could be de-repressed or induced following Meis2 inactivation. Finally, the scRNAseq by Usoskin et al was performed on adult sensory neurons whereas our bulk RNAseq was performed on E18.5 embryos. Thus, because gene expression in developing sensory neurons is well-known to be highly dynamic, it is not expected that the transcriptional signature of sensory neurons subclasses in E18.5 embryo perfectly matches the transcriptional signature of adult subclasses. Finally, we agree that immunostaining for Meis2 together with few selected DEGs would give a better answer on whether they co-localize or not, but our lack of experience with those antibodies together with the lack of financial support for the proposal precludes achieving this pertinent point.

(1.5) The authors identify Gabra1 and Gabra4 as upregulated and Gabrr1 as downregulated genes in MEIS2 cKO animals. Does this reflect a change in GABA-receptor subunit composition in LMTRs?

This is an interesting point. First, in our new analysis, increasing the cut-off to 100 reads excluded Gabrr1 from the DEGs. Based on our results, we cannot conclude whereas Gabra1 and Gabra4 up-regulation reflects a change in GABA receptors composition. However, in the GO term associated to Gabaergic synapse, whereas Gabra1 and Gabra4 were up-regulated the ionotropic glutamate receptor Grid1 was downregulated, rather claiming for an imbalanced GABA/Glutamate transmission. Finally, the increased GABAR expression in the LTMRs might be expected to increase pre-synaptic inhibition on the LTMR synapses onto target neurons in the dorsal horn, thus decreasing synaptic transmission from these neurons into spinal circuits.

1. The authors assessed SA-LTMR innervating Merkel cells in glabrous and hairy skin by IFC staining for neurofilament H and electrophysiological recordings. Due to the small sample size, they pooled recordings, reasoning that nerves that do not successfully innervate Merkel cells (i.e. cKO glabrous skin) do not evoke electrophysiological responses following a touch stimulus.(2.1) It is undoubtedly true that non-innervating nerves will likely not show electrophysiological responses. However, by pooling the recordings of SA-LTMRs from glabrous and hairy skin, the data obtained from the 20% successful recordings of SA-LTMRs from glabrous cKO skin (according to Fig. 4E, upper panel) will be overrepresented and hence lead to a systematic bias. How many recordings were made from the glabrous and hairy skin of each genotype? In case the number of recordings from cKO/glabrous skin is the limiting factor, does the observed difference in vibration threshold hold true when only recordings from hairy skin are compared?

As explained in the text and in our answers to reviewer 1, data for hairy and glabrous SAMs where initially pooled as no differences between them were expected, and next planned electrophysiological experiments were compromised due to the Covid19 pandemic. We are sorry that at this point, we cannot provide additional experiments to clarify this important point.

3. From the IFC images shown in Fig. 6A, it is not clear how the authors quantified branch points and innervated hair follicles.

Branch points correspond to every time a nerve split in 2 or more nerves. Innervated follicles correspond to follicles that are entangled by circumferential and/or lanceolate Nefh+ endings.

4. The quality of the data is very high, but there are several ambiguities and errors in their presentation.

We apologize for this mistake. Figure 1 Supplementary 1 that reports data from Cat walk analysis is now appropriately included in the files.

(4.2) Fig. 3A is confusing and the figure legend just repeats what is already said in the text. What do yellow, blue, and pink represent?

Figure 3 is now fully remade. Legend is now better indicated in Figure 3A. We hope it is now more clear.

(4.3) What genotype do the black, grey, and white boxplots in Fig. 6C Fig. 3 Supplementary 1B correspond to?

The legends were missing for Figure 6C and Figure 3 supplementary 1B. They are now appropriately included.

(4.4) Up- and downregulated genes are assigned differently in Fig. 3 and Fig. 3 Supplementary 2. The figure legend of Fig. 3 Suppl 2 lists panel B as up-regulated genes but the same genes are labeled down-regulated in Fig. 3.

We apologize for this previous mistake. Figure 3 and corresponding supplementary figures have been redone in the new version.

(4.5) Fig. 3E would benefit from a more detailed description. One can easily appreciate that the neurofilament H staining in the cKO sample is different from that of the WT sample but what exactly can be seen here?

We added the following sentence in the results section: “In WT newborn mice, numerous Nefh+ sensory fibers surround all dermal papillae of the hairy skin and footpad of the glabrous skin, whereas in Wnt1Cre::Meis2LoxP/LoxP littermates, very few Nefh+ sensory fibers are present and they poorly innervate the dermal papillae and footpads.“.

(4.6) The figure legend to Fig. 4A is unclear. Does the graph show the sum of all recordings performed? From the text, one would guess that the bars correspond to the cKO samples, but this is not specified. Do the controls correspond to WT, Islet1Cre/+ or a mixture of both? In addition, the graph in the lower panel is labeled % Ab fibers, the figure legend reads % of tap units among Ab fibers.

The graphs show the number of tap units identified among all recorded Aβfibers. Numbers show the number of tap units over the number of recorded fibers. This as been now reformulated in the last version of the manuscript.

(4.7) The abbreviation SAM in figure legends 4F, G is not introduced.

This is now indicated in the figure legend.

(4.8) Readers who are not familiar with the traces above the graphs in 4F and 4G will find a more detailed description helpful.

This is now indicated in the figure legend.

(4.9) Lines 274-275: Does the statement "Finally, consistent with the lack of neuronal loss in Isl1Cre/+::Meis2LoxP/LoxP, the number of recorded fibers were identical in WT and Isl1Cre/+::Meis2LoxP/LoxP." refer to Fig. 4G? This is not specified in the text.

These data were not included in the first version of the manuscript as we though they were not significantly informative. They just indicate the overall numbers of fibers that were recorded in electrophysiological experiments. The sentence has been now removed in the last version of the manuscript to avoid misunderstanding.

(4.10) There is no Fig. 6 supplementary 1.

The typo is now corrected. The corresponding data were in fact in Figure 5 Supplementary 1.

Minor points:Gangfuß et al. report that a patient previously diagnosed with a range of neurological deficits including the diagnosis of severe infantile autism is heterozygous mutant for MEIS2. Although this study links MEIS2 gene function to ASD in the wider sense, adding a few additional references will make the link stronger. Examples are Shimojima et al., Hum Genome Var 2017 or Bae et al., Science 2022.

These two references have been now included in the introduction section of the manuscript.

In some figures (e.g. Fig. 4) the numbering of the panels does not follow the order in which the respective data are mentioned in the text.

Figure 4 is now re-organized so that panels follow the same order as in the results section.